# Net-Trim: Convex Pruning of Deep Neural Networks with Performance Guarantee

**Alireza Aghasi**[*]
Institute for Insight
Georgia State University
IBM TJ Watson
aaghasi@gsu.edu

**Afshin Abdi**
Department of ECE
Georgia Tech
abdi@gatech.edu

**Nam Nguyen**
IBM TJ Watson
nnguyen@us.ibm.com

**Justin Romberg**
Department of ECE
Georgia Tech
jrom@ece.gatech.edu

## Abstract

We introduce and analyze a new technique for model reduction for deep neural networks. While large networks are theoretically capable of learning arbitrarily complex models, overfitting and model redundancy negatively affects the prediction accuracy and model variance. Our Net-Trim algorithm prunes (sparsifies) a trained network layer-wise, removing connections at each layer by solving a convex optimization program. This program seeks a sparse set of weights at each layer that keeps the layer inputs and outputs consistent with the originally trained model. The algorithms and associated analysis are applicable to neural networks operating with the rectified linear unit (ReLU) as the nonlinear activation. We present both parallel and cascade versions of the algorithm. While the latter can achieve slightly simpler models with the same generalization performance, the former can be computed in a distributed manner. In both cases, Net-Trim significantly reduces the number of connections in the network, while also providing enough regularization to slightly reduce the generalization error. We also provide a mathematical analysis of the consistency between the initial network and the retrained model. To analyze the model sample complexity, we derive the general sufficient conditions for the recovery of a sparse transform matrix. For a single layer taking independent Gaussian random vectors of length $N$ as inputs, we show that if the network response can be described using a maximum number of $s$ non-zero weights per node, these weights can be learned from $\mathcal{O}(s \log N)$ samples.

## 1   Introduction

With enough layers, neurons in each layer, and a sufficiently large set of training data, neural networks can learn structure of arbitrary complexity [1]. This model flexibility has made the deep neural network a pioneer machine learning tool over the past decade (see [2] for a comprehensive overview). In practice, multi-layer networks often have more parameters than can be reliably estimated from the amount of data available. This gives the training procedure a certain ambiguity – many different sets of parameter values can model the data equally well, and we risk instabilities due to overfitting. In this paper, we introduce a framework for *sparsifying* networks that have already been trained using standard techniques. This reduction in the number of parameters needed to specify the network makes it more robust and more computationally efficient to implement without sacrificing performance.

---

[*]Corresponding Author

In recent years there has been increasing interest in the mathematical understanding of deep networks. These efforts are mainly in the context of characterizing the minimizers of the underlying cost function [3, 4] and the geometry of the loss function [5]. Recently, the analysis of deep neural networks using compressed sensing tools has been considered in [6], where the distance preservability of feedforward networks at each layer is studied. There are also works on formulating the training of feedforward networks as an optimization problem [7, 8, 9], where the majority of the works approach their understanding of neural networks by sequentially studying individual layers.

Various methods have been proposed to reduce overfitting via regularizing techniques and pruning strategies. These include explicit regularization using $\ell_1$ and $\ell_2$ penalties during training [10, 11], and techniques that randomly remove active connections in the training phase (e.g. Dropout [12] and DropConnect [13]) making them more likely to produce sparse networks. There has also been recent works on explicit network compression (e.g., [14, 15, 16]) to remove the inherent redundancies. In what is perhaps the most closely related work to what is presented below, [14] proposes a pruning scheme that simply truncates small weights of an already trained network, and then re-adjusts the remaining active weights using another round of training. These aforementioned techniques are based on heuristics, and lack general performance guarantees that help understand when and how well they work.

We present a framework, called *Net-Trim*, for pruning the network layer-by-layer that is based on convex optimization. Each layer of the net consists of a linear map followed by a nonlinearity; the algorithms and theory presented below use a rectified linear unit (ReLU) applied point-wise to each output of the linear map. Net-trim works by taking a trained network, and then finding the sparsest set of weights for each layer that keeps the output responses consistent with the initial training. More concisely, if $Y^{(\ell-1)}$ is the input (across the training examples) to layer $\ell$, and $Y^{(\ell)}$ is the output following the ReLU operator, Net-Trim searches for a sparse $W$ such that $Y^{(\ell)} \approx \mathrm{ReLU}(W^\top Y^{(\ell-1)})$. Using the standard $\ell_1$ relaxation for sparsity and the fact that the ReLU function is piecewise linear allows us to perform this search by solving a convex program. In contrast to techniques based on thresholding (such as [14]), Net-Trim does not require multiple other time-consuming training steps after the initial pruning.

Along with making the computations tractable, Net-Trim's convex formulation also allows us to derive theoretical guarantees on how far the retrained model is from the initial model, and establish sample complexity arguments about the number of random samples required to retrain a presumably sparse layer. To the best of our knowledge, Net-Trim is the first pruning scheme with such performance guarantees. In addition, it is easy to modify and adapt to other structural constraints on the weights by adding additional penalty terms or introducing additional convex constraints.

An illustrative example is shown in Figure 1. Here, 200 points in the 2D plane are used to train a binary classifier. The regions corresponding to each class are nested spirals. We fit a classifier using a simple neural network with two hidden layers with fully connected weights, each consisting 200 neurons. Figure 1(b) shows the weighted adjacency matrix between the layers after training, and then again after Net-Trim is applied. With only a negligible change to the overall network response (panel (a) vs panel (d)), Net-Trim is able to prune more than 93% of the links among the neurons, representing a significant model reduction. Even when the neural network is trained using sparsifying weight regularizers (here, Dropout [12] and $\ell_1$ penalty), Net-Trim produces a model which is over 7 times sparser than the initial one, as presented in panel (c). The numerical experiments in Section 6 show that these kinds of results are not limited to toy examples; Net-Trim achieves significant compression ratios on large networks trained on real data sets.

The remainder of the paper is structured as follows. In Section 2, we formally present the network model used in the paper. The proposed pruning schemes, both the parallel and cascade Net-Trim are presented and discussed in Section 3. Section 4 is devoted to the convex analysis of the proposed framework and its sample complexity. The implementation details of the proposed convex scheme are presented in Section 5. Finally, in Section 6, we report some retraining experiments using the Net-Trim and conclude the paper by presenting some general remarks. Along with some extended discussions, the proofs of all of the theoretical statements in the paper are presented as a supplementary note (specifically, §4 of the notes is devoted to the technical proofs).

We very briefly summarize the notation used below. For a matrix $A$, we use $A_{\Gamma_1,:}$ to denote the submatrix formed by restricting the rows of $A$ to the index set $\Gamma_1$. Similarly, $A_{:,\Gamma_2}$ is the submatrix of columns indexed by $\Gamma_2$, and $A_{\Gamma_1,\Gamma_2}$ is formed by extracting both rows and columns. For an

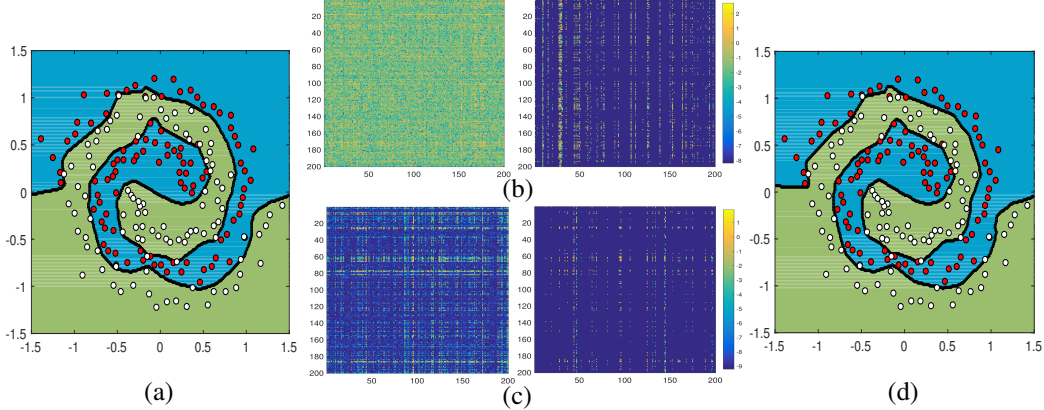

Figure 1: Net-Trim pruning performance; (a) initial trained model; (b) the weighted adjacency matrix relating the two hidden layers before (left) and after (right) the application of Net-Trim; (c) left: the adjacency matrix after training the network with Dropout and $\ell_1$ regularization; right: after retraining via Net-Trim; (d) the retrained classifier

$M \times N$ matrix $\boldsymbol{X}$ with entries $x_{m,n}$, we use[2] $\|\boldsymbol{X}\|_1 \triangleq \sum_{m=1}^{M} \sum_{n=1}^{N} |x_{m,n}|$ and $\|\boldsymbol{X}\|_F$ as the Frobenius norm. For a vector $\boldsymbol{x}$, $\|\boldsymbol{x}\|_0$ is the cardinality of $\boldsymbol{x}$, supp $\boldsymbol{x}$ is the set of indexes with non-zero entries, and supp$^c$ $\boldsymbol{x}$ is the complement set. We will use the notation $\boldsymbol{x}^+$ as shorthand $\max(\boldsymbol{x}, 0)$, where $\max(.,0)$ is applied to vectors and matrices component-wise. Finally, the vertical concatenation of two vectors $\boldsymbol{a}$ and $\boldsymbol{b}$ is denoted by $[\boldsymbol{a}; \boldsymbol{b}]$.

## 2 Feedforward Network Model

In this section, we introduce some notational conventions related to a feedforward network model. We assume that we have $P$ training samples $\boldsymbol{x}_p$, $p = 1, \cdots, P$, where $\boldsymbol{x}_p \in \mathbb{R}^N$ is an input to the network. We stack up these samples into a matrix $\boldsymbol{X} \in \mathbb{R}^{N \times P}$, structured as $\boldsymbol{X} = [\boldsymbol{x}_1, \cdots, \boldsymbol{x}_P]$. Considering $L$ layers for the network, the output of the network at the final layer is denoted by $\boldsymbol{Y}^{(L)} \in \mathbb{R}^{N_L \times P}$, where each column in $\boldsymbol{Y}^{(L)}$ is a response to the corresponding training column in $\boldsymbol{X}$.

The network activations are taken to be rectified linear units. The output of the $\ell$-th layer is $\boldsymbol{Y}^{(\ell)} \in \mathbb{R}^{N_\ell \times P}$, generated by applying the adjoint of the weight matrix $\boldsymbol{W}_\ell \in \mathbb{R}^{N_{\ell-1} \times N_\ell}$ to the output of the previous layer $\boldsymbol{Y}^{(\ell-1)}$ and then applying a component-wise $\max(.,0)$ operation:

$$\boldsymbol{Y}^{(\ell)} = \max\left(\boldsymbol{W}_\ell^\top \boldsymbol{Y}^{(\ell-1)}, 0\right), \qquad \ell = 1, \cdots, L, \tag{1}$$

where $\boldsymbol{Y}^{(0)} = \boldsymbol{X}$ and $N_0 = N$. A trained neural network as outlined in (1) is represented by $\mathcal{TN}(\{\boldsymbol{W}_\ell\}_{\ell=1}^L, \boldsymbol{X})$.

For the sake of theoretical analysis, all the results presented in this paper are stated for *link-normalized* networks, where $\|\boldsymbol{W}_\ell\|_1 = 1$ for every layer $\ell = 1, \cdots, L$. Such presentation is with no loss of generality, as any network in the form of (1) can be converted to its link-normalized version by replacing $\boldsymbol{W}_\ell$ with $\boldsymbol{W}_\ell / \|\boldsymbol{W}_\ell\|_1$, and $\boldsymbol{Y}^{(\ell+1)}$ with $\boldsymbol{Y}^{(\ell+1)} / \prod_{j=0}^\ell \|\boldsymbol{W}_j\|_1$. Since $\max(\alpha x, 0) = \alpha \max(x, 0)$ for $\alpha > 0$, any weight processing on a network of the form (1) can be applied to the link-normalized version and later transferred to the original domain via a suitable scaling.

## 3 Convex Pruning of the Network

Our pruning strategy relies on redesigning the network so that for the same training data each layer outcomes stay more or less close to the initial trained model, while the weights associated with each layer are replaced with sparser versions to reduce the model complexity. Figure 2 presents the main idea, where the complex paths between the layer outcomes are replaced with simple paths. In a sense, if we consider each layer response to the transmitted data as a checkpoint, Net-Trim assures the checkpoints remain roughly the same, while a simpler path between the checkpoints is discovered.

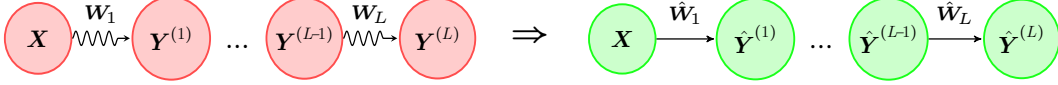

Figure 2: The main retraining idea: keeping the layer outcomes close to the initial trained model while finding a simpler path relating each layer input to the output

Consider the first layer, where $\boldsymbol{X} = [\boldsymbol{x}_1, \cdots, \boldsymbol{x}_P]$ is the layer input, $\boldsymbol{W} = [\boldsymbol{w}_1, \cdots, \boldsymbol{w}_M]$ the layer coefficient matrix, and $\boldsymbol{Y} = [y_{m,p}]$ the layer outcome. We require the new coefficient matrix $\hat{\boldsymbol{W}}$ to be sparse and the new response to be close to $\boldsymbol{Y}$. Using the sum of absolute entries as a proxy to promote sparsity, a natural strategy to retrain the layer is addressing the nonlinear program

$$\hat{\boldsymbol{W}} = \arg\min_{\boldsymbol{U}} \|\boldsymbol{U}\|_1 \quad s.t. \quad \left\|\max\left(\boldsymbol{U}^\top \boldsymbol{X}, 0\right) - \boldsymbol{Y}\right\|_F \le \epsilon. \tag{2}$$

Despite the convex objective, the constraint set in (2) is non-convex. However, we may approximate it with a convex set by imposing $\boldsymbol{Y}$ and $\hat{\boldsymbol{Y}} = \max(\hat{\boldsymbol{W}}^\top \boldsymbol{X}, 0)$ to have similar activation patterns. More specifically, knowing that $y_{m,p}$ is either zero or positive, we enforce the $\max(., 0)$ argument to be negative when $y_{m,p} = 0$, and close to $y_{m,p}$ elsewhere. To present the convex formulation, for $\boldsymbol{V} = [v_{m,p}]$, throughout the paper we use the notation $\boldsymbol{U} \in \mathcal{C}_\epsilon(\boldsymbol{X}, \boldsymbol{Y}, \boldsymbol{V})$ to present the constraint set

$$\begin{cases} \sum\limits_{m,p:\, y_{m,p}>0} \left(\boldsymbol{u}_m^\top \boldsymbol{x}_p - y_{m,p}\right)^2 \le \epsilon^2 \\ \boldsymbol{u}_m^\top \boldsymbol{x}_p \le v_{m,p} \qquad\quad m, p:\, y_{m,p} = 0 \end{cases}. \tag{3}$$

Based on this definition, a convex proxy to (2) is

$$\hat{\boldsymbol{W}} = \arg\min_{\boldsymbol{U}} \ \|\boldsymbol{U}\|_1 \quad s.t. \quad \boldsymbol{U} \in \mathcal{C}_\epsilon(\boldsymbol{X}, \boldsymbol{Y}, \boldsymbol{0}). \tag{4}$$

Basically, depending on the value of $y_{m,p}$, a different constraint is imposed on $\boldsymbol{u}_m^\top \boldsymbol{x}_p$ to emulate the ReLU operation. As a first observation towards establishing a retraining framework, we show that the solution of (4) is consistent with the desired constraint in (2), as follows.

**Proposition 1.** *Let $\hat{\boldsymbol{W}}$ be the solution to (4). For $\hat{\boldsymbol{Y}} = \max(\hat{\boldsymbol{W}}^\top \boldsymbol{X}, 0)$ being the retrained layer response, $\|\hat{\boldsymbol{Y}} - \boldsymbol{Y}\|_F \le \epsilon$.*

### 3.1 Parallel and Cascade Net-Trim

Based on the above exploratory, we propose two schemes to retrain a neural network; one explores a computationally distributable nature and the other proposes a cascading scheme to retrain the layers sequentially. The general idea which originates from the relaxation in (4) is referred to as the Net-Trim, specified by the parallel or cascade nature.

The parallel Net-Trim is a straightforward application of the convex program (4) to each layer in the network. Basically, each layer is processed independently based on the initial model input and output, without taking into account the retraining result from the previous layer. Specifically, denoting $\boldsymbol{Y}^{(\ell-1)}$ and $\boldsymbol{Y}^{(\ell)}$ as the input and output of the $\ell$-th layer of the initial trained neural network (see equation (1)), we propose to relearn the coefficient matrix $\boldsymbol{W}_\ell$ via the convex program

$$\hat{\boldsymbol{W}}_\ell = \arg\min_{\boldsymbol{U}} \|\boldsymbol{U}\|_1 \quad s.t. \quad \boldsymbol{U} \in \mathcal{C}_\epsilon\left(\boldsymbol{Y}^{(\ell-1)}, \boldsymbol{Y}^{(\ell)}, \boldsymbol{0}\right). \tag{5}$$

The optimization in (5) can be independently applied to every layer in the network and hence computationally distributable. Algorithm 1 presents the pseudocode for the parallel Net-Trim. In this pseudocode, we use $\mathtt{TRIM}(\boldsymbol{X}, \boldsymbol{Y}, \boldsymbol{V}, \epsilon)$ as a function which returns the solution to a program like (4) with the constraint $\boldsymbol{U} \in \mathcal{C}_\epsilon(\boldsymbol{X}, \boldsymbol{Y}, \boldsymbol{V})$.

With reference to the constraint in (5), if we only retrain the $\ell$-th layer, the output of the retrained layer is in the $\epsilon$-neighborhood of that before retraining. However, when all the layers are retrained through (5), an immediate question would be whether the retrained network produces an output which is controllably close to the initially trained model. In the following theorem, we show that the retrained error does not blow up across the layers and remains a multiple of $\epsilon$.

**Theorem 1.** *Let $\mathcal{TN}(\{W_\ell\}_{\ell=1}^L, X)$ be a link-normalized trained network with layer outcomes $Y^{(\ell)}$ described by (1). Form the retrained network $\mathcal{TN}(\{\hat{W}_\ell\}_{\ell=1}^L, X)$ by solving the convex programs (5), with $\epsilon = \epsilon_\ell$ at each layer. Then the retrained layer outcomes $\hat{Y}^{(\ell)} = \max(\hat{W}_\ell^\top \hat{Y}^{(\ell-1)}, 0)$ obey $\|\hat{Y}^{(\ell)} - Y^{(\ell)}\|_F \le \sum_{j=1}^\ell \epsilon_j$.*

When all the layers are retrained with a fixed parameter $\epsilon$ (as in Algorithm 1), a corollary of the theorem above would bound the overall discrepancy as $\|\hat{Y}^{(L)} - Y^{(L)}\|_F \le L\epsilon$.

In a cascade Net-Trim, unlike the parallel scheme where each layer is retrained independently, the outcome of a retrained layer is probed into the program retraining the next layer. More specifically, having the first layer processed via (4), one would ideally seek to address (5) with the modified constraint $U \in \mathcal{C}_\epsilon(\hat{Y}^{(\ell-1)}, Y^{(\ell)}, 0)$ to retrain the subsequent layers. However, as detailed in §1 of the supplementary note, such program is not necessarily feasible and needs to be sufficiently slacked to warrant feasibility. In this regard, for every subsequent layer, $\ell = 2, \cdots, L$, the retrained weighting matrix, $\hat{W}_\ell$, is obtained via

$$\min_U \|U\|_1 \quad s.t. \quad U \in \mathcal{C}_{\epsilon_\ell}\left(\hat{Y}^{(\ell-1)}, Y^{(\ell)}, W_\ell^\top \hat{Y}^{(\ell-1)}\right), \tag{6}$$

where for $W_\ell = [w_{\ell,1}, \cdots, w_{\ell,N_\ell}]$ and $\gamma_\ell \ge 1$,

$$\epsilon_\ell^2 = \gamma_\ell \sum_{m,p:\, y_{m,p}^{(\ell)}>0} \left(w_{\ell,m}^\top \hat{y}_p^{(\ell-1)} - y_{m,p}^{(\ell)}\right)^2. \tag{7}$$

The constants $\gamma_\ell \ge 1$ (referred to as the *inflation rates*) are free parameters, which control the sparsity of the resulting matrices. In the following theorem, we prove that the outcome of the retrained network produced by Algorithm 2 is close to that of the network before retraining.

**Theorem 2.** *Let $\mathcal{TN}(\{W_\ell\}_{\ell=1}^L, X)$ be a link-normalized trained network with layer outcomes $Y^{(\ell)}$. Form the retrained network $\mathcal{TN}(\{\hat{W}_\ell\}_{\ell=1}^L, X)$ by solving (5) for the first layer and (6) for the subsequent layers with $\epsilon_\ell$ as in (7), $\hat{Y}^{(\ell)} = \max(\hat{W}_\ell^\top \hat{Y}^{(\ell-1)}, 0)$, $\hat{Y}^{(1)} = \max(\hat{W}_1^\top X, 0)$ and $\gamma_\ell \ge 1$. Then the outputs $\hat{Y}^{(\ell)}$ of the retrained network will obey $\|\hat{Y}^{(\ell)} - Y^{(\ell)}\|_F \le \epsilon_1 (\prod_{j=2}^\ell \gamma_j)^{\frac{1}{2}}$.*

Algorithm 2 presents the pseudo-code to implement the cascade Net-Trim for a link normalized network with $\epsilon_1 = \epsilon$ and a constant inflation rate, $\gamma$, across all the layers. In such case, a corollary of Theorem 2 bounds the network overall discrepancy as $\|\hat{Y}^{(L)} - Y^{(L)}\|_F \le \gamma^{\frac{(L-1)}{2}} \epsilon$.

We would like to note that focusing on a link-normalized network is only for the sake of presenting the theoretical results in a more compact form. In practice, such conversion is not necessary and to retrain layer $\ell$ in the parallel Net-Trim we can take $\epsilon = \epsilon_r \|Y^{(\ell)}\|_F$ and use $\epsilon = \epsilon_r \|Y^{(1)}\|_F$ for the cascade case, where $\epsilon_r$ plays a similar role as $\epsilon$ for a link-normalized network. Moreover, as detailed in §2 of the supplementary note, Theorems 1 and 2 identically apply to the practical networks that follow (1) for the first $L - 1$ layers and skip an activation at the last layer.

---

| **Algorithm 1** Parallel Net-Trim | **Algorithm 2** Cascade Net-Trim |
|---|---|
| | 1: **Input:** $X$, $\epsilon > 0$, $\gamma > 1$ and normalized $W_1, \cdots, W_L$ |
| 1: **Input:** $X$, $\epsilon > 0$, and normalized $W_1, \cdots, W_L$ | 2: $Y \leftarrow \max(W_1^\top X, 0)$ |
| 2: $Y^{(0)} \leftarrow X$ | 3: $\hat{W}_1 \leftarrow \text{TRIM}(X, Y, 0, \epsilon)$ |
|     % generating initial layer outcomes: | 4: $\hat{Y} \leftarrow \max(\hat{W}_1^\top X, 0)$ |
| 3: **for** $\ell = 1, \cdots, L$ **do** | 5: **for** $\ell = 2, \cdots, L$ **do** |
| 4:    $Y^{(\ell)} \leftarrow \max(W_\ell^\top Y^{(\ell-1)}, 0)$ | 6:    $Y \leftarrow \max(W_\ell^\top Y, 0)$ |
| 5: **end for** | 7:    $\epsilon \leftarrow (\gamma \sum_{m,p:y_{m,p}>0}(w_{\ell,m}^\top \hat{y}_p - y_{m,p})^2)^{1/2}$ |
|     % retraining: |     % $w_{\ell,m}$ is the $m$-th column of $W_\ell$ |
| 6: **for all** $\ell = 1, \cdots, L$ **do** | 8:    $\hat{W}_\ell \leftarrow \text{TRIM}(\hat{Y}, Y, W_\ell^\top \hat{Y}, \epsilon)$ |
| 7:    $\hat{W}_\ell \leftarrow \text{TRIM}(Y^{(\ell-1)}, Y^{(\ell)}, 0, \epsilon)$ | 9:    $\hat{Y} \leftarrow \max(\hat{W}_\ell^\top \hat{Y}, 0)$ |
| 8: **end for** | 10: **end for** |
| 9: **Output:** $\hat{W}_1, \cdots, \hat{W}_L$ | 11: **Output:** $\hat{W}_1, \cdots, \hat{W}_L$ |

# 4 Convex Analysis and Sample Complexity

In this section, we derive a sampling theorem for a single-layer, redundant network. Here, there are many sets of weights that can induce the observed outputs given then input vectors. This scenario might arise when the number of training samples used to train a (large) network is small (smaller than the network degrees of freedom). We will show that when the inputs into the layers are independent Gaussian random vectors, if there are sparse set of weights that can generate the output, then with high probability, the Net-Trim program in (4) will find them.

As noted above, in the case of a redundant layer, for a given input $\boldsymbol{X}$ and output $\boldsymbol{Y}$, the relation $\boldsymbol{Y} = \max(\boldsymbol{W}^\top \boldsymbol{X}, 0)$ can be established via more than one $\boldsymbol{W}$. In this case, we hope to find a sparse $\boldsymbol{W}$ by setting $\epsilon = 0$ in (4). For this value of $\epsilon$, our central convex program decouples into $M$ convex programs, each searching for the $m$-th column in $\hat{\boldsymbol{W}}$:

$$\hat{\boldsymbol{w}}_m = \arg\min_{\boldsymbol{w}} \|\boldsymbol{w}\|_1 \ \ s.t. \ \begin{cases} \boldsymbol{w}^\top \boldsymbol{x}_p = y_{m,p} & p : y_{m,p} > 0 \\ \boldsymbol{w}^\top \boldsymbol{x}_p \le 0 & p : y_{m,p} = 0 \end{cases}. \tag{8}$$

By dropping the $m$ index and introducing the slack variable $\boldsymbol{s}$, program (8) can be cast as

$$\min_{\boldsymbol{w},\boldsymbol{s}} \ \|\boldsymbol{w}\|_1 \qquad s.t. \qquad \tilde{\boldsymbol{X}} \begin{bmatrix} \boldsymbol{w} \\ \boldsymbol{s} \end{bmatrix} = \boldsymbol{y}, \quad \boldsymbol{s} \le \boldsymbol{0}, \tag{9}$$

where

$$\tilde{\boldsymbol{X}} = \begin{bmatrix} \boldsymbol{X}_{:,\Omega}^\top & \boldsymbol{0} \\ \boldsymbol{X}_{:,\Omega^c}^\top & -\boldsymbol{I} \end{bmatrix}, \quad \boldsymbol{y} = \begin{bmatrix} \boldsymbol{y}_\Omega \\ \boldsymbol{0} \end{bmatrix},$$

and $\Omega = \{p : y_p > 0\}$. For a general $\tilde{\boldsymbol{X}}$, not necessarily structured as above, the following result states the sufficient conditions under which a sparse pair $(\boldsymbol{w}^*, \boldsymbol{s}^*)$ is the unique minimizer to (9).

**Proposition 2.** *Consider a pair $(\boldsymbol{w}^*, \boldsymbol{s}^*) \in (\mathbb{R}^{n_1}, \mathbb{R}^{n_2})$, which is feasible for the convex program (9). If there exists a vector $\boldsymbol{\Lambda} = [\Lambda_\ell] \in \mathbb{R}^{n_1+n_2}$ in the range of $\tilde{\boldsymbol{X}}^\top$ with entries satisfying*

$$\begin{cases} -1 < \Lambda_\ell < 1 & \ell \in supp^c \ \boldsymbol{w}^* \\ 0 < \Lambda_{n_1+\ell} & \ell \in supp^c \ \boldsymbol{s}^* \end{cases}, \quad \begin{cases} \Lambda_\ell = sign(w_\ell^*) & \ell \in supp \ \boldsymbol{w}^* \\ \Lambda_{n_1+\ell} = 0 & \ell \in supp \ \boldsymbol{s}^* \end{cases} \tag{10}$$

*and for $\tilde{\Gamma} = supp \ \boldsymbol{w}^* \cup \{n_1 + supp \ \boldsymbol{s}^*\}$ the restricted matrix $\tilde{\boldsymbol{X}}_{:,\tilde{\Gamma}}$ is full column rank, then the pair $(\boldsymbol{w}^*, \boldsymbol{s}^*)$ is the unique solution to (9).*

The proposed optimality result can be related to the unique identification of a sparse $\boldsymbol{w}^*$ from rectified observations of the form $\boldsymbol{y} = \max(\boldsymbol{X}^\top \boldsymbol{w}^*, 0)$. Clearly, the structure of the feature matrix $\boldsymbol{X}$ plays the key role here, and the construction of the dual certificate stated in Proposition 2 entirely relies on this. As an insightful case, we show that when $\boldsymbol{X}$ is a *Gaussian matrix* (that is, the elements of $\boldsymbol{X}$ are i.i.d values drawn from a standard normal distribution), for sufficiently large number of samples, the dual certificate can be constructed. As a result, we can warrant that learning $\boldsymbol{w}^*$ can be performed with much fewer samples than the layer degrees of freedom.

**Theorem 3.** *Let $\boldsymbol{w}^* \in \mathbb{R}^N$ be an arbitrary s-sparse vector, $\boldsymbol{X} \in \mathbb{R}^{N \times P}$ a Gaussian matrix representing the samples and $\mu > 1$ a fixed value. Given $P = (11s + 7)\mu \log N$ observations of the type $\boldsymbol{y} = \max(\boldsymbol{X}^\top \boldsymbol{w}^*, 0)$, with probability exceeding $1 - N^{1-\mu}$ the vector $\boldsymbol{w}^*$ can be learned exactly through (8).*

The standard Gaussian assumption for the feature matrix $\boldsymbol{X}$ allows us to relate the number of training samples to the number of active links in a layer. Such feature structure could be a realistic assumption for the first layer of the neural network. As reflected in the proof of Theorem 3, because of the dependence of the set $\Omega$ to the entries in $\boldsymbol{X}$, we need to take a totally nontrivial analysis path than the standard concentration of measure arguments for the sum of independent random matrices. In fact, the proof requires establishing concentration bounds for the sum of dependent random matrices.

While we focused on each column of $\boldsymbol{W}^*$ individually, for the observations $\boldsymbol{Y} = \max(\boldsymbol{W}^{*\top} \boldsymbol{X}, 0)$, using the union bound, an exact identification of $\boldsymbol{W}^*$ can be warranted as a corollary of Theorem 3.

**Corollary 1.** *Consider an arbitrary matrix $\boldsymbol{W}^* = [\boldsymbol{w}_1^*, \cdots, \boldsymbol{w}_M^*] \in \mathbb{R}^{N \times M}$, where $s_m = \|\boldsymbol{w}_m^*\|_0$, and $0 < s_m \le s_{\max}$ for $m = 1, \cdots, M$. For $\boldsymbol{X} \in \mathbb{R}^{N \times P}$ being a Gaussian matrix, set $\boldsymbol{Y} = \max(\boldsymbol{W}^{*\top} \boldsymbol{X}, 0)$. If $\mu > (1 + \log_N M)$ and $P = (11s_{\max} + 7)\mu \log N$, for $\epsilon = 0$, $\boldsymbol{W}^*$ can be accurately learned through (4) with probability exceeding $1 - \sum_{m=1}^M N^{1-\mu \frac{11s_{\max}+7}{11s_m+7}}$.*

It can be shown that for the network model in (1), probing the network with an i.i.d sample matrix $\boldsymbol{X}$ would generate subgaussian random matrices with independent columns as the subsequent layer outcomes. Under certain well conditioning of the input covariance matrix of each layer, results similar to Theorem 3 are extendable to the subsequent layers. While such results are left for a more extended presentation of the work, Theorem 3 is brought here as a good reference for the general performance of the proposed retraining scheme and the associated analysis theme.

## 5 Implementing the Convex Program

If the quadratic constraint in (3) is brought to the objective via a regularization parameter $\lambda$, the resulting convex program decouples into $M$ smaller programs of the form

$$\hat{\boldsymbol{w}}_m = \arg\min_{\boldsymbol{u}} \ \|\boldsymbol{u}\|_1 + \lambda \sum_{p:\, y_{m,p}>0} \left(\boldsymbol{u}^\top \boldsymbol{x}_p - y_{m,p}\right)^2 \quad s.t. \quad \boldsymbol{u}^\top \boldsymbol{x}_p \le v_{m,p}, \ \text{for} \ \ p:\, y_{m,p}=0, \quad (11)$$

each recovering a column of $\hat{\boldsymbol{W}}$. Such decoupling of the regularized form is computationally attractive, since it makes the trimming task extremely distributable among parallel processing units by recovering each column of $\hat{\boldsymbol{W}}$ on a separate unit. Addressing the original constrained form (4) in a fast and scalable way requires using more complicated techniques, which is left to a more extended presentation of the work.

We can formulate the program in a standard form by introducing the index sets

$$\Omega_m = \{p : y_{m,p} > 0\}, \quad \Omega_m^c = \{p : y_{m,p} = 0\}.$$

Denoting the $m$-th row of $\boldsymbol{Y}$ by $\boldsymbol{y}_m^\top$ and the $m$-th row of $\boldsymbol{V}$ by $\boldsymbol{v}_m^\top$, one can equivalently rewrite (11) in terms of $\boldsymbol{u}$ as

$$\min_{\boldsymbol{u}} \ \|\boldsymbol{u}\|_1 + \boldsymbol{u}^\top \boldsymbol{Q}_m \boldsymbol{u} + 2\boldsymbol{q}_m^\top \boldsymbol{u} \quad s.t. \quad \boldsymbol{P}_m \boldsymbol{u} \le \boldsymbol{c}_m, \quad (12)$$

where

$$\boldsymbol{Q}_m = \lambda \boldsymbol{X}_{:,\Omega_m} \boldsymbol{X}_{:,\Omega_m}^\top, \quad \boldsymbol{q}_m = -\lambda \boldsymbol{X}_{:,\Omega_m} \boldsymbol{y}_{m\Omega_m} = -\lambda \boldsymbol{X} \boldsymbol{y}_m, \quad \boldsymbol{P}_m = \boldsymbol{X}_{:,\Omega_m^c}^\top, \quad \boldsymbol{c}_m = \boldsymbol{v}_{m\Omega_m^c}. \quad (13)$$

The $\ell_1$ term in the objective of (12) can be converted into a linear term by defining a new vector $\tilde{\boldsymbol{u}} = [\boldsymbol{u}^+; -\boldsymbol{u}^-]$, where $\boldsymbol{u}^- = \min(\boldsymbol{u}, 0)$. This variable change naturally yields

$$\boldsymbol{u} = [\boldsymbol{I}, -\boldsymbol{I}]\tilde{\boldsymbol{u}}, \quad \|\boldsymbol{u}\|_1 = \boldsymbol{1}^\top \tilde{\boldsymbol{u}}.$$

The convex program (13) is now cast as the standard quadratic program

$$\min_{\tilde{\boldsymbol{u}}} \ \tilde{\boldsymbol{u}}^\top \tilde{\boldsymbol{Q}}_m \tilde{\boldsymbol{u}} + (\boldsymbol{1} + 2\tilde{\boldsymbol{q}}_m)^\top \tilde{\boldsymbol{u}} \quad s.t. \quad \begin{bmatrix} \tilde{\boldsymbol{P}}_m \\ -\boldsymbol{I} \end{bmatrix} \tilde{\boldsymbol{u}} \le \begin{bmatrix} \boldsymbol{c}_m \\ \boldsymbol{0} \end{bmatrix}, \quad (14)$$

where

$$\tilde{\boldsymbol{Q}}_m = \begin{bmatrix} 1 & -1 \\ -1 & 1 \end{bmatrix} \otimes \boldsymbol{Q}_m, \quad \tilde{\boldsymbol{q}}_m = \begin{bmatrix} \boldsymbol{q}_m \\ -\boldsymbol{q}_m \end{bmatrix}, \quad \tilde{\boldsymbol{P}}_m = [\boldsymbol{P}_m \quad -\boldsymbol{P}_m].$$

Once $\tilde{\boldsymbol{u}}_m^*$, the solution to (14) is found, the solution to (11) can be recovered via $\hat{\boldsymbol{w}}_m = [\boldsymbol{I}, -\boldsymbol{I}]\tilde{\boldsymbol{u}}_m^*$.

Aside from the variety of convex solvers that can be used to address (14), we are specifically interested in using the alternating direction method of multipliers (ADMM). In fact the main motivation to translate (11) into (14) is the availability of ADMM implementations for problems in the form of (14) that are reasonably fast and scalable (e.g., see [17]). The authors have made the implementation publicly available online[3].

## 6 Experiments and Discussions

Aside from the major technical contribution of the paper in providing a theoretical understanding of the Net-Trim pruning process, in this section we present some experiments to highlight its performance against the state of the art techniques.

The first set of experiments associated with the example presented in the introduction (classification of 2D points on nested spirals) compares the Net-Trim pruning power against the standard pruning strategies of $\ell_1$ regularization and Dropout. The experiments demonstrate how Net-Trim can significantly improve the pruning level of a given network and produce simpler and more understandable networks. We also compare the cascade Net-Trim against the parallel scheme. As could be expected, for a fixed level of discrepancy between the initial and retrained models, the cascade scheme is capable of producing sparser networks. However, the computational distributability of the parallel scheme makes it a more favorable approach for large scale and big data problems. Due to the space limitation, these experiments are moved to §3 of the supplementary note.

We next apply Net-Trim to the problem of classifying hand-written digits of the mixed national institute of standards and technology (MNIST) dataset. The set contains 60,000 training samples and 10,000 test instances. To examine different settings, we consider 6 networks: NN2-10K, which is a $784 \cdot 300 \cdot 300 \cdot 10$ network (two hidden layers of 300 nodes) and trained with 10,000 samples; NN3-30K, a $784 \cdot 300 \cdot 500 \cdot 300 \cdot 10$ network trained with 30,000 samples; and NN3-60K, a $784 \cdot 300 \cdot 1000 \cdot 300 \cdot 10$ network trained with 60,000 samples. We also consider CNN-10K, CNN-30K and CNN-60K which are topologically identical convolutional networks trained with 10,000, 30,000 and 60,000 samples, respectively. The convolutional networks contain two convolutional layers composed of 32 filters of size $5 \times 5 \times 1$ for the first layer and $5 \times 5 \times 32$ for the second layer, both followed by max pooling and a fully connected layer of 512 neurons. While the linearity of the convolution allows using the Net-Trim for the associated layers, here we merely consider retraining the fully connected layers.

To address the Net-Trim convex program, we use the regularized form outlined in Section 5, which is fully capable of parallel processing. For our largest problem (associated with the fully connected layer in CNN-60K), retraining each column takes less than 20 seconds and distributing the independent jobs among a cluster of processing units (in our case 64) or using a GPU reduces the overall retraining of a layer to few minutes.

Table 1 summarize the retraining experiments. Panel (a) corresponds to the Net-Trim operating in a low discrepancy mode (smaller $\epsilon$), while in panel (b) we explore more sparsity by allowing larger discrepancies. Each neural network is trained three times with different initialization seeds and average quantities are reported. In these tables, the first row corresponds to the test accuracy of the initial models. The second row reports the overall pruning rate and the third row reports the overall discrepancy between the initial and Net-Trim retrained models. We also compare the results with the work by Han, Pool, Tran and Dally (HPTD) [14]. The basic idea in [14] is to truncate the small weights across the network and perform another round of training on the active weights. The forth row reports the test accuracy after applying Net-Trim. To make a fair comparison in applying the HPTD, we impose the same number of weights to be truncated in the HPTD technique. The accuracy of the model after this truncation is presented on the fifth row. Rows six and seven present the test accuracy of Net-Trim and HPTD after a fine training process (optional for Net-Trim).

An immediate observation is the close test error of Net-Trim compared to the initial trained models (row four vs row one). We can observe from the second and third rows of the two tables that allowing more discrepancy (larger $\epsilon$) increases the pruning level. We can also observe that the basic Net-Trim process (row four) in many scenarios beats the HPTD (row seven), and if we allow a fine training step after the Net-Trim (row six), in all the scenarios a better test accuracy is achieved.

A serious problem with the HPTD is the early minima trapping (EMT). When we simply truncate the layer transfer matrices, ignoring their actual contribution to the network, the error introduced can be very large (row five), and using this biased pattern as an initialization for the fine training can produce poor local minima solutions with large errors. The EMT blocks in the table correspond to the scenarios where all three random seeds failed to generate acceptable results for this approach. In the experiments where Net-Trim was followed by an additional fine training step, this was never an issue, since the Net-Trim outcome is already a good model solution.

In Figure 3(a), we visualize $\hat{W}_1$ after the Net-Trim process. We observe 28 bands (MNIST images are 28×28), where the zero columns represent the boundary pixels with the least image information. It is noteworthy that such interpretable result is achieved using the Net-Trim with no post or pre-processes. A similar outcome of HPTD is depicted in panel (b). As a matter of fact, the authors present a similar visualization as panel (a) in [14], which is the result of applying the HPTD process iteratively and going through the retraining step many times. Such path certainly produces a lot of processing load and lacks any type of confidence on being a convergent procedure.

Table 1: The test accuracy of different models before and after Net-Trim (NT) and HPTD [14]. Without a fine training (FT) step, Net-Trim produces pruned networks in the majority of cases more accurate than HPTD and with no risk of poor local minima. Adding an additional FT step makes Net-Trim consistently prominent

|  | NN2-10K | NN3-30K | NN3-60K | CNN-10K | CNN-30K | CNN-60K |
|---|---|---|---|---|---|---|
| Init. Mod. Acc. (%) | 95.59 | 97.58 | 98.18 | 98.37 | 99.11 | 99.25 |
| Total Pruning (%) | 40.86 | 30.69 | 29.38 | 43.91 | 39.11 | 45.74 |
| NT Overall Disc. (%) | 1.98 | 1.31 | 1.77 | 1.22 | 0.75 | 0.55 |
| NT No FT Acc. (%) | 95.47 | 97.55 | 98.1 | 98.31 | 99.15 | 99.25 |
| HPTD No FT Acc. (%) | 9.3 | 10.34 | 8.92 | 19.17 | 55.92 | 30.17 |
| NT + FT Acc. (%) | 95.85 | 97.67 | 98.12 | 98.35 | 99.21 | 99.33 |
| HPTD + FT Acc. (%) | 93.56 | 97.32 | EMT | 98.16 | EMT | EMT |

(a)

|  | NN2-10K | NN3-30K | NN3-60K | CNN-10K | CNN-30K | CNN-60K |
|---|---|---|---|---|---|---|
| Init. Mod. Acc. (%) | 95.59 | 97.58 | 98.18 | 98.37 | 99.11 | 99.25 |
| Total Pruning (%) | 75.87 | 75.82 | 77.40 | 76.18 | 77.63 | 81.62 |
| NT Overall Disc. (%) | 4.95 | 11.01 | 11.47 | 3.65 | 5.32 | 8.93 |
| NT No FT Acc. (%) | 94.92 | 95.97 | 97.35 | 97.91 | 99.08 | 98.96 |
| HPTD No FT Acc. (%) | 8.97 | 10.1 | 8.92 | 31.18 | 73.36 | 46.84 |
| NT + FT Acc. (%) | 95.89 | 97.69 | 98.19 | 98.40 | 99.17 | 99.26 |
| HPTD + FT Acc. (%) | 95.61 | EMT | 97.96 | EMT | 99.01 | 99.06 |

(b)

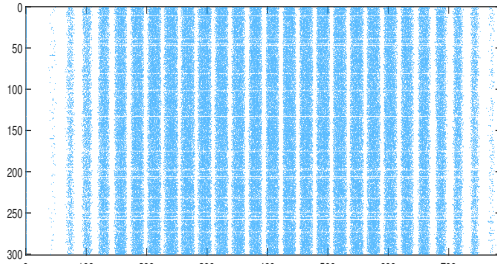
(a)

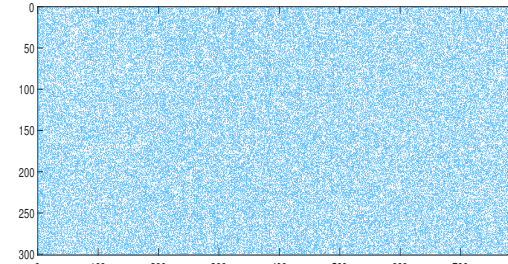
(b)

Figure 3: Visualization of $\hat{W}_1$ in NN3-60K; (a) Net-Trim output; (b) standard HPTD

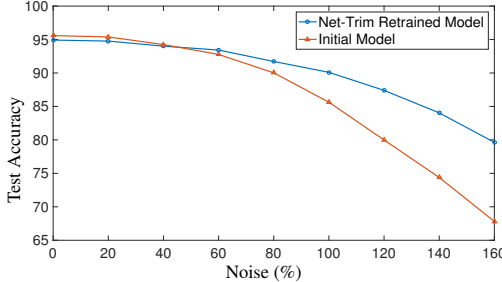
(a)

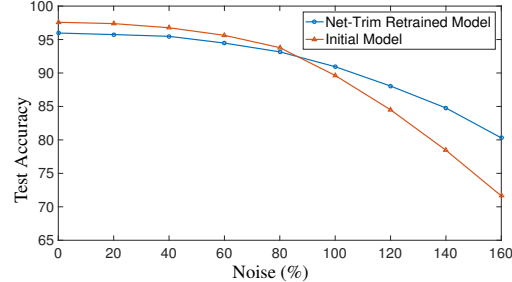
(b)

Figure 4: Noise robustness of initial and retrained networks; (a) NN2-10K; (b) NN3-30K

Also, for a deeper understanding of the robustness Net-Trim adds to the models, in Figure 4 we have plotted the classification accuracy of the initial and retrained models against the level of added noise to the test data (ranging from 0 to 160%). The Net-Trim improvement in accuracy becomes more noticeable as the noise level in the data increases. Basically, as expected, reducing the model complexity makes the network more robust to outliers and noisy samples. It is also interesting to note that the NN3-30K initial model in panel (b), which is trained with more data, presents robustness to a larger level of noise compared to NN2-10K in panel (a). However, the retrained models behave rather similarly (blue curves) indicating the saving that can be achieved in the number of training samples via Net-Trim.

In fact, Net-Trim can be particularly useful when the number of training samples is limited. While overfitting is likely to occur in such scenarios, Net-Trim reduces the complexity of the model by setting a significant portion of weights at each layer to zero, yet maintaining the model consistency. This capability can also be viewed from a different perspective, that Net-Trim simplifies the process of determining the network size. In other words, if the network used at the training phase is oversized, Net-Trim can reduce its size to an order matching the data. Finally, aside from the theoretical and practical contribution that Net-Trim brings to the understanding of deep neural network, the idea can be easily generalized to retraining schemes with other regularizers (e.g., the use of ridge or elastic net type regularizers) or other structural constraint about the network.

## Footnotes

[2]The notation $\|\boldsymbol{X}\|_1$ should not be confused with the matrix induced $\ell_1$ norm

[3]The code for the regularized Net-Trim implementation using the ADMM scheme can be accessed online at: `https://github.com/DNNToolBox/Net-Trim-v1`

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
