[Supplementary Material]

# Supplementary Note
# Net-Trim: Convex Pruning of Deep Neural Networks with Performance Guarantee

Alireza Aghasi, Afshin Abdi, Nam Nguyen and Justin Romberg*

**Abstract**

The material presented in this document is supplementary to the manuscript presented at the NIPS 2017. The document contains extended discussions on some of the material presented in the paper, an extended experiment highlighting the Net-Trim performance and the technical proofs of the statements presented in the paper.

## Contents

*Contact: `aaghasi@gsu.edu`

# 1 Maintaining the Feasibility for the Cascade Net-Trim

As pointed out in the text, in the case of cascade Net-Trim, the proposed convex program needs to be slacked to prevent an infeasibility issue. To better explain the mechanics, consider starting the cascade process by retraining the first layer as

$$\hat{\boldsymbol{W}}_1 = \arg\min_{\boldsymbol{U}} \ \|\boldsymbol{U}\|_1 \quad s.t. \quad \boldsymbol{U} \in \mathcal{C}_{\epsilon_1}\left(\boldsymbol{X}, \boldsymbol{Y}^{(1)}, \boldsymbol{0}\right). \tag{S.1}$$

Setting $\hat{\boldsymbol{Y}}^{(1)} = \max(\hat{\boldsymbol{W}}_1^\top \boldsymbol{X}, 0)$ to be the outcome of the retrained layer, to retrain the second layer, we ideally would like to address a similar program as (S.1) with $\hat{\boldsymbol{Y}}^{(1)}$ as the input and $\boldsymbol{Y}^{(2)}$ being the output reference, i.e.,

$$\min_{\boldsymbol{U}} \ \|\boldsymbol{U}\|_1 \quad s.t. \quad \boldsymbol{U} \in \mathcal{C}_{\epsilon_2}\left(\hat{\boldsymbol{Y}}^{(1)}, \boldsymbol{Y}^{(2)}, \boldsymbol{0}\right). \tag{S.2}$$

However, there is no guarantee that program (S.2) is feasible, that is, there exists a matrix $\boldsymbol{W} = [\boldsymbol{w}_1, \cdots, \boldsymbol{w}_{N_2}]$ such that

$$\begin{cases} \sum\limits_{m,p:\ y_{m,p}^{(2)}>0} (\boldsymbol{w}_m^\top \hat{\boldsymbol{y}}_p^{(1)} - y_{m,p}^{(2)})^2 \leq \epsilon_2^2 \\ \boldsymbol{w}_m^\top \hat{\boldsymbol{y}}_p^{(1)} \leq 0 \qquad m, p: \ y_{m,p}^{(2)} = 0 \end{cases}. \tag{S.3}$$

If instead of $\hat{\boldsymbol{Y}}^{(1)}$ the constraint set (S.2) was parameterized by $\boldsymbol{Y}^{(1)}$, a natural feasible point would have been $\boldsymbol{W}_2$. Now that $\hat{\boldsymbol{Y}}^{(1)}$ is a perturbed version of $\boldsymbol{Y}^{(1)}$, the constraint set needs to be slacked to maintain the feasibility of $\boldsymbol{W}_2$. In this context, one may easily verify that

$$\boldsymbol{W}_2 \in \mathcal{C}_{\epsilon_2}\left(\hat{\boldsymbol{Y}}^{(1)}, \boldsymbol{Y}^{(2)}, \boldsymbol{W}_2^\top \hat{\boldsymbol{Y}}^{(1)}\right) \tag{S.4}$$

as long as $\epsilon_2^2 \geq \sum\limits_{m,p:\ y_{m,p}^{(2)}>0} \left(\boldsymbol{w}_{2,m}^\top \hat{\boldsymbol{y}}_p^{(1)} - y_{m,p}^{(2)}\right)^2$, where $\boldsymbol{w}_{2,m}$ is the $m$-th column of $\boldsymbol{W}_2$. Basically the constraint set in (S.4) is a slacked version of the constraint set in (S.3), where the right hand side quantities in the corresponding inequalities are sufficiently extended to maintain the feasibility of $\boldsymbol{W}_2$. A similar argument applies to all subsequent layers justifying formulations (6) and (7) of the paper.

# 2 Retraining the Last Layer

Commonly, the last layer in a neural network is not subject to an activation function and a standard linear model applies, i.e., $\boldsymbol{Y}^{(L)} = \boldsymbol{W}_L^\top \boldsymbol{Y}^{(L-1)}$. This linear outcome may be directly exploited for regression purposes or pass through a soft-max function to produce the scores for a classification task.

In this case, to retrain the layer we simply need to seek a sparse weight matrix under the constraint that the linear outcomes stay close before and after retraining. More specifically,

$$\hat{\boldsymbol{W}}_L = \underset{\boldsymbol{U}}{\arg\min} \ \|\boldsymbol{U}\|_1 \quad s.t. \quad \left\|\boldsymbol{U}^\top \boldsymbol{Y}^{(L-1)} - \boldsymbol{Y}^{(L)}\right\|_F \le \epsilon_L. \tag{S.5}$$

In the case of cascade Net-Trim,

$$\hat{\boldsymbol{W}}_L = \underset{\boldsymbol{U}}{\arg\min} \ \|\boldsymbol{U}\|_1 \quad s.t. \quad \left\|\boldsymbol{U}^\top \hat{\boldsymbol{Y}}^{(L-1)} - \boldsymbol{Y}^{(L)}\right\|_F \le \epsilon_L, \tag{S.6}$$

and the feasibility of the program is established for

$$\epsilon_L^2 = \gamma_L \left\|\boldsymbol{W}_L^\top \hat{\boldsymbol{Y}}^{(L-1)} - \boldsymbol{Y}^{(L)}\right\|_F^2, \qquad \gamma \ge 1. \tag{S.7}$$

It can be shown that the results stated earlier in Theorems 1 and 2 regarding the overall discrepancy of the network generalize to a network with linear activation at the last layer.

**Proposition S.1.** *Consider a link-normalized network $\mathcal{TN}(\{\boldsymbol{W}_\ell\}_{\ell=1}^L, \boldsymbol{X})$, where a standard linear model applies to the last layer.*

*(a) If the first $L-1$ layers are retrained according to the process stated in Theorem 1 and the last layer is retrained through (S.5), then*

$$\left\|\hat{\boldsymbol{Y}}^{(L)} - \boldsymbol{Y}^{(L)}\right\|_F \le \sum_{\ell=1}^L \epsilon_j.$$

*(b) If the first $L-1$ layers are retrained according to the process stated in Theorem 2 and the last layer is retrained through (S.6) and (S.7), then*

$$\left\|\hat{\boldsymbol{Y}}^{(L)} - \boldsymbol{Y}^{(L)}\right\|_F \le \epsilon_1 \sqrt{\prod_{j=2}^L \gamma_j}.$$

*Proof.* For part (a), from Theorem 1 we have $\|\hat{\boldsymbol{Y}}^{(L-1)} - \boldsymbol{Y}^{(L-1)}\|_F \le \sum_{\ell=1}^{L-1} \epsilon_\ell$. Furthermore,

$$
\begin{aligned}
\left\|\hat{\boldsymbol{Y}}^{(L)} - \boldsymbol{Y}^{(L)}\right\|_F &= \left\|\hat{\boldsymbol{W}}_L^\top \hat{\boldsymbol{Y}}^{(L-1)} - \boldsymbol{Y}^{(L)}\right\|_F \\
&\le \left\|\hat{\boldsymbol{W}}_L^\top \hat{\boldsymbol{Y}}^{(L-1)} - \hat{\boldsymbol{W}}_L^\top \boldsymbol{Y}^{(L-1)}\right\|_F + \left\|\hat{\boldsymbol{W}}_L^\top \boldsymbol{Y}^{(L-1)} - \boldsymbol{Y}^{(L)}\right\|_F \\
&\le \left\|\hat{\boldsymbol{W}}_L^\top\right\|_F \left\|\hat{\boldsymbol{Y}}^{(L-1)} - \boldsymbol{Y}^{(L-1)}\right\|_F + \epsilon_L \\
&\le \sum_{\ell=1}^L \epsilon_\ell,
\end{aligned}
$$

where for the last inequality we used a similar chain of inequalities as (S.19).

To prove part (b), for the first $L-1$ layers, $\|\hat{\boldsymbol{Y}}^{(L-1)} - \boldsymbol{Y}^{(L-1)}\|_F \le \epsilon_1 \sqrt{\prod_{\ell=1}^{L-1} \gamma_\ell}$, and therefore

$$
\begin{aligned}
\left\| \hat{\boldsymbol{Y}}^{(L)} - \boldsymbol{Y}^{(L)} \right\|_F &= \left\| \hat{\boldsymbol{W}}_L^\top \hat{\boldsymbol{Y}}^{(L-1)} - \boldsymbol{Y}^{(L)} \right\|_F \\
&\le \sqrt{\gamma_L} \left\| \boldsymbol{W}_L^\top \hat{\boldsymbol{Y}}^{(L-1)} - \boldsymbol{W}_L^\top \boldsymbol{Y}^{(L-1)} \right\|_F \\
&\le \sqrt{\gamma_L} \left\| \boldsymbol{W}_L \right\|_F \left\| \hat{\boldsymbol{Y}}^{(L-1)} - \boldsymbol{Y}^{(L-1)} \right\|_F \\
&\le \sqrt{\gamma_L} \left\| \hat{\boldsymbol{Y}}^{(L-1)} - \boldsymbol{Y}^{(L-1)} \right\|_F \\
&\le \epsilon_1 \sqrt{\prod_{\ell=1}^{L} \gamma_\ell}.
\end{aligned}
$$

Here, the first inequality is thanks to (S.6) and (S.7).

$\square$

While the cascade Net-Trim is designed in way that infeasibility is never an issue, one can take a slight risk of infeasibility in retraining the last layer to further reduce the overall discrepancy. More specifically, if the value of $\epsilon_L$ in (S.6) is replaced with $\kappa\epsilon_L$ for some $\kappa \in (0,1)$, we may reduce the overall discrepancy by the factor $\kappa$, without altering the sparsity pattern of the first $L-1$ layers. It is however clear that in this case there is no guarantee that program (S.6) remains feasible and multiple trials may be needed to tune $\kappa$. We will refer to $\kappa$ as the risk coefficient.

# 3  Additional Experiment: Classification of Data on Nested Spirals

For a better demonstration of the Net-trim performance in terms of model reduction, mean accuracy and cascade vs. parallel retraining frameworks, here we focus on a low dimensional dataset. We specifically look into the classification of two set of points lying on nested spirals as shown in Figure S.1(a). The dataset is embedded into the H2O package [1, 2] and publicly available along with the module.

As an initial experiment, we consider a network of size 2·200·200·2, which indicates the use of two hidden layers of 200 neurons each, i.e., $\boldsymbol{W}_1 \in \mathbb{R}^{2\times200}$, $\boldsymbol{W}_2 \in \mathbb{R}^{200\times200}$ and $\boldsymbol{W}_3 \in \mathbb{R}^{200\times2}$. After training the model, a contour plot of the soft-max outcome, indicating the classifier, is depicted in Figure S.1(b). We apply the cascade Net-Trim for $\epsilon = 0.01 \times \|\boldsymbol{Y}^{(1)}\|_F$ (the network is not link normalized), $\gamma = 1.1$ and the final risk coefficient $\kappa = 0.35$ (see Section 2 of this

**Figure S.1:** Classifying two set of data points on nested spirals; (a) the points corresponding to each class with different colors; (b) the soft-max contour (0.5 level-set) representing the neural net classifier; (c) the classifier after applying the Net-Trim (d) a plot of the network weights corresponding to the last layer, before (on the left side) and after (on the right side) retraining

supplementary note for the definition of $\kappa$). To evaluate the difference between the network output before and after retraining, we define the relative discrepancy

$$\epsilon_{rd} = \frac{\|\boldsymbol{Z} - \hat{\boldsymbol{Z}}\|_F}{\|\boldsymbol{Z}\|_F}, \tag{S.8}$$

where $\boldsymbol{Z} = \boldsymbol{W}_3 \boldsymbol{Y}^{(2)}$ and $\hat{\boldsymbol{Z}} = \hat{\boldsymbol{W}}_3 \hat{\boldsymbol{Y}}^{(2)}$ are the network outcomes before the soft-max operation. In this case $\epsilon_{rd} = 0.046$. The classifier after retraining is presented in Figure S.1(c), which shows minor difference with the original classifier in panel (b). The number of nonzero elements in $\boldsymbol{W}_1, \boldsymbol{W}_2$ and $\boldsymbol{W}_3$ are 397, 39770 and 399, respectively. After retraining, the active entries in $\hat{\boldsymbol{W}}_1, \hat{\boldsymbol{W}}_2$ and $\hat{\boldsymbol{W}}_3$ reduce to 362, 2663 and 131 elements, respectively. Basically, at the expense of a slight model discrepancy, a significant reduction in the model complexity is achieved. Figures 1(b) and S.1(d) compare the cardinalities of the second and third layer weights before and after retraining.

**Figure S.2:** Sparsity ratio as a function of overall network relative mismatch for the cascade (first row) and parallel (second row) schemes

As a second experiment, we train the neural network with Dropout and $\ell_1$ penalty to produce a readily simple model. The number of nonzero elements in $\boldsymbol{W}_1, \boldsymbol{W}_2$ and $\boldsymbol{W}_3$ turn out to be 319, 6554 and 304, respectively. Using a similar $\epsilon$ as the first experiment, we apply the cascade Net-Trim, which produces a retrained model with $\epsilon_{rd} = 0.0183$ (the classifiers are visually identical and not shown here). The number of active entries in $\hat{\boldsymbol{W}}_1, \hat{\boldsymbol{W}}_2$ and $\hat{\boldsymbol{W}}_3$ are 189, 929 and 84, respectively. Despite the use of model reduction tools (Dropout and $\ell_1$ penalty) in the training phase, the Net-Trim yet zeros out a large portion of the weights in the retraining phase. The second layer weight-matrix densities before and after retraining are visually comparable in Figure 1(c).

We next perform a more extensive experiment to evaluate the performance of the cascade Net-Trim against the parallel version. Using the spiral data, we train three networks each with two hidden layers of sizes $50 \cdot 50$, $75 \cdot 75$ and $100 \cdot 100$. For the parallel retraining, we fix a value of $\epsilon$, retrain each model 20 times and record the mean layer sparsity across these experiments (the averaging among 20 experiments is to remove the bias of local minima in the training phase). A similar process is repeated for the cascade case, where we consistently use $\gamma = 1.1$ and $\kappa = 1$. We can sweep the values of $\epsilon$ in a range to generate a class of curves relating the network relative discrepancy to each layer mean sparsity ratio, as presented in Figure S.2. Here, sparsity ratio refers to the ratio of active elements to the total number of elements in the weight matrix.

A natural observation from the decreasing curves is allowing more discrepancy leads to

more level of sparsity. We also observe that for a constant discrepancy $\epsilon_{rd}$, the cascade Net-Trim is capable of generating rather sparser networks. The contrast in sparsity is more apparent in the third layer (panel (c) vs. panel (f)). We would like to note that using $\kappa < 1$ makes the contrast even more tangible, however for the purpose of long-run simulations, here we chose $\kappa = 1$ to avoid any possible infeasibility interruptions. Finally, an interesting observation is the rather dense retrained matrices associated with the first layer. Apparently, less pruning takes place at the first layer to maximally bring the information and data structure into the network.

In Table S.1 we have listed some retraining scenarios for networks of different sizes trained with Dropout. Across all the experiments, we have used the cascade Net-Trim to retrain the networks and chosen $\epsilon$ small enough to warrant an overall relative discrepancy below 0.02. On the right side of the table, the number of active elements for each layer is reported, which indicates the significant model reduction for a negligible discrepancy.

**Table S.1:** Number of active elements within each layer, before and after Net-Trim for a network trained with Dropout

| Network Size | Trained Network | | | Net-Trim Retrained Network | | |
|---|---|---|---|---|---|---|
| | Layer 1 | Layer 2 | Layer 3 | Layer 1 | Layer 2 | Layer 3 |
| $2 \cdot 50 \cdot 50 \cdot 2$ | 99 | 2483 | 100 | 98 | 467 | 54 |
| $2 \cdot 75 \cdot 75 \cdot 2$ | 149 | 5594 | 150 | 149 | 710 | 72 |
| $2 \cdot 125 \cdot 125 \cdot 2$ | 250 | 15529 | 250 | 247 | 3477 | 96 |
| $2 \cdot 175 \cdot 175 \cdot 2$ | 349 | 30395 | 350 | 348 | 1743 | 116 |
| $2 \cdot 200 \cdot 200 \cdot 2$ | 400 | 39668 | 399 | 399 | 1991 | 113 |

**Table S.2:** Number of active elements within each layer, before and after Net-Trim for a network trained with Dropout and an $\ell_1$-penalty

| Network Size | Trained Network | | | Net-Trim Retrained Network | | |
|---|---|---|---|---|---|---|
| | Layer 1 | Layer 2 | Layer 3 | Layer 1 | Layer 2 | Layer 3 |
| $2 \cdot 50 \cdot 50 \cdot 2$ | 58 | 1604 | 95 | 54 | 342 | 46 |
| $2 \cdot 75 \cdot 75 \cdot 2$ | 96 | 2867 | 135 | 90 | 651 | 62 |
| $2 \cdot 125 \cdot 125 \cdot 2$ | 126 | 5316 | 226 | 95 | 751 | 60 |
| $2 \cdot 175 \cdot 175 \cdot 2$ | 171 | 9580 | 320 | 136 | 906 | 61 |
| $2 \cdot 200 \cdot 200 \cdot 2$ | 134 | 8700 | 382 | 109 | 606 | 70 |

Table S.2 reports another set of sample experiments, where Dropout and $\ell_1$ penalty are simultaneously employed in the training phase to prune the network. Going through a similar cascade retraining, while keeping $\epsilon_{rd}$ below 0.02, we have reported the level of additional model

**Figure S.3:** The weight histogram of the middle layer before and after retraining; (a) the middle layer histogram of a $2 \cdot 50 \cdot 50 \cdot 2$ network trained with Dropout (left) vs. the histogram after Net-Trim (right); (b) similar plots as panel (a) for a $2 \cdot 200 \cdot 200 \cdot 2$ network

reduction that can be achieved. Basically, the Net-Trim post processing module uses the trained model (regardless of how it is trained) to further reduce its complexity. A comparison of the network weight histograms before and after retraining may better highlight the Net-Trim performance. Figure S.3 compares the middle layer weight histograms for a pair of experiments reported in Table S.1.

# 4    Technical Proofs

## 4.1    Proof of Proposition 1

If $\hat{\boldsymbol{W}} = [\hat{\boldsymbol{w}}_1, \cdots, \hat{\boldsymbol{w}}_M]$ is a solution to (4), then the feasibility of the solution requires

$$\sum_{m,p \,:\, y_{m,p} > 0} (\hat{\boldsymbol{w}}_m^\top \boldsymbol{x}_p - y_{m,p})^2 \le \epsilon^2 \tag{S.9}$$

and

$$\hat{\boldsymbol{w}}_m^\top \boldsymbol{x}_p \le 0 \qquad \text{if} \quad y_{m,p} = 0. \tag{S.10}$$

Consider $\hat{\boldsymbol{Y}} = [\hat{y}_{m,p}]$, then

$$\left\|\boldsymbol{Y} - \hat{\boldsymbol{Y}}\right\|_F^2 = \sum_{m=1}^{M}\sum_{p=1}^{P}(y_{m,p} - \hat{y}_{m,p})^2$$

$$= \sum_{m,p\,:\,y_{m,p}>0}(y_{m,p} - \hat{y}_{m,p})^2 + \sum_{m,p\,:\,y_{m,p}=0}(y_{m,p} - \hat{y}_{m,p})^2$$

$$= \sum_{m,p\,:\,y_{m,p}>0}\left(y_{m,p} - (\hat{\boldsymbol{w}}_m^\top \boldsymbol{x}_p)^+\right)^2$$

$$= \sum_{m,p\,:\,y_{m,p}>0,\hat{\boldsymbol{w}}_m^\top \boldsymbol{x}_p>0}\left(y_{m,p} - \hat{\boldsymbol{w}}_m^\top \boldsymbol{x}_p\right)^2 + \sum_{m,p\,:\,y_{m,p}>0,\hat{\boldsymbol{w}}_m^\top \boldsymbol{x}_p\leq 0} y_{m,p}^2. \tag{S.11}$$

Here since $y_{m,p} \geq 0$, the second equality is partitioned into two summations separated by the values of $y_{m,p}$ being zero or strictly greater than zero. The second resulting sum vanishes in the third equality since from (S.10), $\hat{\boldsymbol{y}}_{m,p} = \max(\hat{\boldsymbol{w}}_m^\top \boldsymbol{x}_p, 0) = 0$ when $\boldsymbol{y}_{m,p} = 0$. For the second term in (S.11) we use the basic algebraic identity

$$\sum_{m,p\,:\,y_{m,p}>0,\hat{\boldsymbol{w}}_m^\top \boldsymbol{x}_p\leq 0} y_{m,p}^2 = \sum_{m,p\,:\,y_{m,p}>0,\hat{\boldsymbol{w}}_m^\top \boldsymbol{x}_p\leq 0}\left(y_{m,p} - \hat{\boldsymbol{w}}_m^\top \boldsymbol{x}_p\right)^2 + 2y_{m,p}(\hat{\boldsymbol{w}}_m^\top \boldsymbol{x}_p) - \left(\hat{\boldsymbol{w}}_m^\top \boldsymbol{x}_p\right)^2. \tag{S.12}$$

Combining (S.12) and (S.11) results in

$$\left\|\boldsymbol{Y} - \hat{\boldsymbol{Y}}\right\|_F^2 = \sum_{m,p\,:\,y_{m,p}>0}\left(y_{m,p} - \hat{\boldsymbol{w}}_m^\top \boldsymbol{x}_p\right)^2 + \sum_{m,p\,:\,y_{m,p}>0,\hat{\boldsymbol{w}}_m^\top \boldsymbol{x}_p\leq 0} 2y_{m,p}(\hat{\boldsymbol{w}}_m^\top \boldsymbol{x}_p) - \left(\hat{\boldsymbol{w}}_m^\top \boldsymbol{x}_p\right)^2. \tag{S.13}$$

From (S.9), the first sum in (S.13) is upper bounded by $\epsilon^2$. In addition,

$$2y_{m,p}(\hat{\boldsymbol{w}}_m^\top \boldsymbol{x}_p) - \left(\hat{\boldsymbol{w}}_m^\top \boldsymbol{x}_p\right)^2 \leq 0,$$

when $y_{m,p} > 0$ and $\hat{\boldsymbol{w}}_m^\top \boldsymbol{x}_p \leq 0$, which together yield $\|\boldsymbol{Y} - \hat{\boldsymbol{Y}}\|_F^2 \leq \epsilon^2$ as expected.

## 4.2   Proof of Theorem 1

We prove the theorem by induction. For $\ell = 1$, the claim holds as a direct result of Proposition 1. Now suppose the claim holds up to the $(\ell - 1)$-th layer,

$$\left\|\hat{\boldsymbol{Y}}^{(\ell-1)} - \boldsymbol{Y}^{(\ell-1)}\right\|_F \leq \sum_{j=1}^{\ell-1}\varepsilon_j, \tag{S.14}$$

we show that for the $\ell$-th layer, $\|\hat{\boldsymbol{Y}}^{(\ell)} - \boldsymbol{Y}^{(\ell)}\|_F \leq \sum_{j=1}^{\ell}\epsilon_j$. The outcome of the $\ell$-th layer before and after retraining obeys

$$y_{m,p}^{(\ell)} = \left(\boldsymbol{w}_m^\top \boldsymbol{y}_p^{(\ell-1)}\right)^+ \qquad \text{and} \qquad \hat{y}_{m,p}^{(\ell)} = \left(\hat{\boldsymbol{w}}_m^\top \hat{\boldsymbol{y}}_p^{(\ell-1)}\right)^+, \tag{S.15}$$

where $y_{m,p}^{(\ell)}$ and $\hat{y}_{m,p}^{(\ell)}$ are entries of $\boldsymbol{Y}^{(\ell)}$ and $\hat{\boldsymbol{Y}}^{(\ell)}$, the $m$-th columns of $\boldsymbol{W}_\ell$ and $\hat{\boldsymbol{W}}_\ell$ are denoted by $\boldsymbol{w}_m$ and $\hat{\boldsymbol{w}}_m$ (we have dropped the $\ell$ subscripts in the column notation for brevity), and the $p$-th columns of $\boldsymbol{Y}^{(\ell-1)}$ and $\hat{\boldsymbol{Y}}^{(\ell-1)}$ are denoted by $\boldsymbol{y}_p^{(\ell-1)}$ and $\hat{\boldsymbol{y}}_p^{(\ell-1)}$. We also define the quantities

$$\tilde{y}_{m,p}^{(\ell)} = \left( \hat{\boldsymbol{w}}_m^\top \boldsymbol{y}_p^{(\ell-1)} \right)^+,$$

which form a matrix $\tilde{\boldsymbol{Y}}^{(\ell)}$. From Proposition 1, we have

$$\left\| \tilde{\boldsymbol{Y}}^{(\ell)} - \boldsymbol{Y}^{(\ell)} \right\|_F \leq \epsilon_\ell. \tag{S.16}$$

On the other hand,

$$
\begin{aligned}
\hat{y}_{m,p}^{(\ell)} &= \left( \hat{\boldsymbol{w}}_m^\top \hat{\boldsymbol{y}}_p^{(\ell-1)} \right)^+ \\
&= \left( \hat{\boldsymbol{w}}_m^\top \boldsymbol{y}_p^{(\ell-1)} + \hat{\boldsymbol{w}}_m^\top \left( \hat{\boldsymbol{y}}_p^{(\ell-1)} - \boldsymbol{y}_p^{(\ell-1)} \right) \right)^+ \\
&\leq \left( \hat{\boldsymbol{w}}_m^\top \boldsymbol{y}_p^{(\ell-1)} \right)^+ + \left( \hat{\boldsymbol{w}}_m^\top \left( \hat{\boldsymbol{y}}_p^{(\ell-1)} - \boldsymbol{y}_p^{(\ell-1)} \right) \right)^+ \\
&\leq \tilde{y}_{m,p}^{(\ell)} + \left| \hat{\boldsymbol{w}}_m^\top \left( \hat{\boldsymbol{y}}_p^{(\ell-1)} - \boldsymbol{y}_p^{(\ell-1)} \right) \right|,
\end{aligned}
\tag{S.17}
$$

where in the last two inequalities we used the sub-additivity of the $\max(.,0)$ function and the inequality $\max(x,0) \leq |x|$. In a similar fashion we have

$$
\begin{aligned}
\tilde{y}_{m,p}^{(\ell)} &= \left( \hat{\boldsymbol{w}}_m^\top \boldsymbol{y}_p^{(\ell-1)} \right)^+ \\
&\leq \left( \hat{\boldsymbol{w}}_m^\top \hat{\boldsymbol{y}}_p^{(\ell-1)} \right)^+ + \left( \hat{\boldsymbol{w}}_m^\top \left( \boldsymbol{y}_p^{(\ell-1)} - \hat{\boldsymbol{y}}_p^{(\ell-1)} \right) \right)^+ \\
&\leq \hat{y}_{m,p}^{(\ell)} + \left| \hat{\boldsymbol{w}}_m^\top \left( \hat{\boldsymbol{y}}_p^{(\ell-1)} - \boldsymbol{y}_p^{(\ell-1)} \right) \right|,
\end{aligned}
$$

which together with (S.17) asserts that $|\hat{y}_{m,p}^{(\ell)} - \tilde{y}_{m,p}^{(\ell)}| \leq |\hat{\boldsymbol{w}}_m^\top (\hat{\boldsymbol{y}}_p^{(\ell-1)} - \boldsymbol{y}_p^{(\ell-1)})|$ or

$$\left\| \hat{\boldsymbol{Y}}^{(\ell)} - \tilde{\boldsymbol{Y}}^{(\ell)} \right\|_F \leq \left\| \hat{\boldsymbol{W}}_\ell^\top \left( \hat{\boldsymbol{Y}}^{(\ell-1)} - \boldsymbol{Y}^{(\ell-1)} \right) \right\|_F \leq \left\| \hat{\boldsymbol{W}}_\ell \right\|_F \left\| \hat{\boldsymbol{Y}}^{(\ell-1)} - \boldsymbol{Y}^{(\ell-1)} \right\|_F. \tag{S.18}$$

As $\hat{\boldsymbol{W}}_\ell$ is the minimizer of (5) and $\boldsymbol{W}_\ell$ is a feasible point (i.e., $\boldsymbol{W}_\ell \in \mathcal{C}_{\epsilon_\ell}(\boldsymbol{Y}^{(\ell-1)}, \boldsymbol{Y}^{(\ell)}, \boldsymbol{0})$), we have

$$\|\hat{\boldsymbol{W}}_\ell\|_F \leq \|\hat{\boldsymbol{W}}_\ell\|_1 \leq \|\boldsymbol{W}_\ell\|_1 = 1, \tag{S.19}$$

which with reference to (S.18) yields

$$\left\| \hat{\boldsymbol{Y}}^{(\ell)} - \tilde{\boldsymbol{Y}}^{(\ell)} \right\|_F \leq \left\| \hat{\boldsymbol{Y}}^{(\ell-1)} - \boldsymbol{Y}^{(\ell-1)} \right\|_F \leq \sum_{j=1}^{\ell-1} \varepsilon_j.$$

Finally, the induction proof is completed by applying the triangle inequality and then using (S.16),

$$\left\|\hat{\boldsymbol{Y}}^{(\ell)} - \boldsymbol{Y}^{(\ell)}\right\|_F \le \left\|\hat{\boldsymbol{Y}}^{(\ell)} - \tilde{\boldsymbol{Y}}^{(\ell)}\right\|_F + \left\|\tilde{\boldsymbol{Y}}^{(\ell)} - \boldsymbol{Y}^{(\ell)}\right\|_F \le \sum_{j=1}^{\ell} \varepsilon_j.$$

## 4.3 Proof of Theorem 2

For $\ell \ge 2$ we relate the upper-bound of $\|\hat{\boldsymbol{Y}}^{(\ell)} - \boldsymbol{Y}^{(\ell)}\|_F$ to $\|\hat{\boldsymbol{Y}}^{(\ell-1)} - \boldsymbol{Y}^{(\ell-1)}\|_F$. By the construction of the network:

$$\left\|\hat{\boldsymbol{Y}}^{(\ell)} - \boldsymbol{Y}^{(\ell)}\right\|_F^2 = \sum_{m=1}^{N_\ell} \sum_{p=1}^{P} \left(\left(\hat{\boldsymbol{w}}_m^\top \hat{\boldsymbol{y}}_p^{(\ell-1)}\right)^+ - \left(\boldsymbol{w}_m^\top \boldsymbol{y}_p^{(\ell-1)}\right)^+\right)^2$$

$$= \sum_{m,p\,:\,y_{m,p}^{(\ell)}>0} \left(\left(\hat{\boldsymbol{w}}_m^\top \hat{\boldsymbol{y}}_p^{(\ell-1)}\right)^+ - \boldsymbol{w}_m^\top \boldsymbol{y}_p^{(\ell-1)}\right)^2 + \sum_{m,p\,:\,y_{m,p}^{(\ell)}=0} \left(\left(\hat{\boldsymbol{w}}_m^\top \hat{\boldsymbol{y}}_p^{(\ell-1)}\right)^+\right)^2, \quad \text{(S.20)}$$

where the $m$-th columns of $\boldsymbol{W}_\ell$ and $\hat{\boldsymbol{W}}_\ell$ are denoted by $\boldsymbol{w}_m$ and $\hat{\boldsymbol{w}}_m$, respectively (we have dropped the $\ell$ subscripts in the column notation for brevity), and the $p$-th columns of $\boldsymbol{Y}^{(\ell)}$ and $\hat{\boldsymbol{Y}}^{(\ell)}$ are denoted by $\boldsymbol{y}_p^{(\ell-1)}$ and $\hat{\boldsymbol{y}}_p^{(\ell-1)}$. Also, as before $y_{m,p}^{(\ell)} = (\boldsymbol{w}_m^\top \boldsymbol{y}_p^{(\ell-1)})^+$. For the first term in (S.20) we have

$$\sum_{m,p\,:\,y_{m,p}^{(\ell)}>0} \left(\left(\hat{\boldsymbol{w}}_m^\top \hat{\boldsymbol{y}}_p^{(\ell-1)}\right)^+ - \boldsymbol{w}_m^\top \boldsymbol{y}_p^{(\ell-1)}\right)^2 = \sum_{m,p\,:\,y_{m,p}^{(\ell)}>0,\,\hat{\boldsymbol{w}}_m^\top \hat{\boldsymbol{y}}_p^{(\ell-1)}>0} \left(\hat{\boldsymbol{w}}_m^\top \hat{\boldsymbol{y}}_p^{(\ell-1)} - \boldsymbol{w}_m^\top \boldsymbol{y}_p^{(\ell-1)}\right)^2$$

$$+ \sum_{m,p\,:\,y_{m,p}^{(\ell)}>0,\,\hat{\boldsymbol{w}}_m^\top \hat{\boldsymbol{y}}_p^{(\ell-1)}\le 0} \left(\boldsymbol{w}_m^\top \boldsymbol{y}_p^{(\ell-1)}\right)^2. \quad \text{(S.21)}$$

However, for the elements of the set $\{(m,p) : y_{m,p}^{(\ell)} > 0,\ \hat{\boldsymbol{w}}_m^\top \hat{\boldsymbol{y}}_p^{(\ell-1)} \le 0\}$:

$$\left(\boldsymbol{w}_m^\top \boldsymbol{y}_p^{(\ell-1)}\right)^2 = \left(\hat{\boldsymbol{w}}_m^\top \hat{\boldsymbol{y}}_p^{(\ell-1)} - \boldsymbol{w}_m^\top \boldsymbol{y}_p^{(\ell-1)}\right)^2 + 2\left(\hat{\boldsymbol{w}}_m^\top \hat{\boldsymbol{y}}_p^{(\ell-1)}\right)\left(\boldsymbol{w}_m^\top \boldsymbol{y}_p^{(\ell-1)}\right) - \left(\hat{\boldsymbol{w}}_m^\top \hat{\boldsymbol{y}}_p^{(\ell-1)}\right)^2$$

$$\le \left(\hat{\boldsymbol{w}}_m^\top \hat{\boldsymbol{y}}_p^{(\ell-1)} - \boldsymbol{w}_m^\top \boldsymbol{y}_p^{(\ell-1)}\right)^2,$$

using which in (S.21) yields

$$\sum_{m,p\,:\,y_{m,p}^{(\ell)}>0} \left(\left(\hat{\boldsymbol{w}}_m^\top \hat{\boldsymbol{y}}_p^{(\ell-1)}\right)^+ - \boldsymbol{w}_m^\top \boldsymbol{y}_p^{(\ell-1)}\right)^2 \le \sum_{m,p\,:\,y_{m,p}^{(\ell)}>0} \left(\hat{\boldsymbol{w}}_m^\top \hat{\boldsymbol{y}}_p^{(\ell-1)} - \boldsymbol{w}_m^\top \boldsymbol{y}_p^{(\ell-1)}\right)^2$$

$$\le \gamma_\ell \sum_{m,p\,:\,y_{m,p}^{(\ell)}>0} \left(\boldsymbol{w}_m^\top \hat{\boldsymbol{y}}_p^{(\ell-1)} - \boldsymbol{w}_m^\top \boldsymbol{y}_p^{(\ell-1)}\right)^2 \quad \text{(S.22)}$$

$$= \epsilon_\ell^2.$$

Here, the second inequality is a direct result of the feasibility condition

$$\hat{\boldsymbol{W}}_\ell \in \mathcal{C}_{\epsilon_\ell}\left(\hat{\boldsymbol{Y}}^{(\ell-1)}, \boldsymbol{Y}^{(\ell)}, \boldsymbol{W}_\ell\hat{\boldsymbol{Y}}^{(\ell-1)}\right). \tag{S.23}$$

A second outcome of the feasibility is

$$\hat{\boldsymbol{w}}_m^\top\hat{\boldsymbol{y}}_p^{(\ell-1)} \le \boldsymbol{w}_m^\top\hat{\boldsymbol{y}}_p^{(\ell-1)}, \tag{S.24}$$

for any pair $(m,p)$ that obeys $y_{m,p}^{(\ell)} = 0$ (or equivalently, $\boldsymbol{w}_m^\top\boldsymbol{y}_p^{(\ell-1)} \le 0$). We can apply $\max(.,0)$ (as an increasing and positive function) to both sides of (S.24) and use it to bound the second term in (S.20) as follows:

$$\sum_{m,p\,:\,y_{m,p}^{(\ell)}=0} \left(\left(\hat{\boldsymbol{w}}_m^\top\hat{\boldsymbol{y}}_p^{(\ell-1)}\right)^+\right)^2 \le \sum_{m,p\,:\,y_{m,p}^{(\ell)}=0} \left(\left(\boldsymbol{w}_m^\top\hat{\boldsymbol{y}}_p^{(\ell-1)}\right)^+\right)^2$$

$$= \sum_{m,p\,:\,y_{m,p}^{(\ell)}=0} \left(\left(\boldsymbol{w}_m^\top\boldsymbol{y}_p^{(\ell-1)} + \boldsymbol{w}_m^\top\hat{\boldsymbol{y}}_p^{(\ell-1)} - \boldsymbol{w}_m^\top\boldsymbol{y}_p^{(\ell-1)}\right)^+\right)^2$$

$$\le \sum_{m,p\,:\,y_{m,p}^{(\ell)}=0} \left(\left(\boldsymbol{w}_m^\top\boldsymbol{y}_p^{(\ell-1)}\right)^+ + \left(\boldsymbol{w}_m^\top\hat{\boldsymbol{y}}_p^{(\ell-1)} - \boldsymbol{w}_m^\top\boldsymbol{y}_p^{(\ell-1)}\right)^+\right)^2$$

$$= \sum_{m,p\,:\,y_{m,p}^{(\ell)}=0} \left(\left(\boldsymbol{w}_m^\top\hat{\boldsymbol{y}}_p^{(\ell-1)} - \boldsymbol{w}_m^\top\boldsymbol{y}_p^{(\ell-1)}\right)^+\right)^2$$

$$\le \sum_{m,p\,:\,y_{m,p}^{(\ell)}=0} \left(\boldsymbol{w}_m^\top\hat{\boldsymbol{y}}_p^{(\ell-1)} - \boldsymbol{w}_m^\top\boldsymbol{y}_p^{(\ell-1)}\right)^2. \tag{S.25}$$

The first and second terms in (S.20) are bounded via (S.22) and (S.25) and therefore

$$\left\|\hat{\boldsymbol{Y}}^{(\ell)} - \boldsymbol{Y}^{(\ell)}\right\|_F^2 \le \gamma_\ell \sum_{m,p\,:\,y_{m,p}^{(\ell)}>0} \left(\boldsymbol{w}_m^\top\hat{\boldsymbol{y}}_p^{(\ell-1)} - \boldsymbol{w}_m^\top\boldsymbol{y}_p^{(\ell-1)}\right)^2 + \sum_{m,p\,:\,y_{m,p}^{(\ell)}=0} \left(\boldsymbol{w}_m^\top\hat{\boldsymbol{y}}_p^{(\ell-1)} - \boldsymbol{w}_m^\top\boldsymbol{y}_p^{(\ell-1)}\right)^2$$

$$\le \gamma_\ell \sum_{m,p} \left(\boldsymbol{w}_m^\top\hat{\boldsymbol{y}}_p^{(\ell-1)} - \boldsymbol{w}_m^\top\boldsymbol{y}_p^{(\ell-1)}\right)^2$$

$$= \gamma_\ell \left\|\boldsymbol{W}_\ell^\top\left(\hat{\boldsymbol{Y}}^{(\ell-1)} - \boldsymbol{Y}^{(\ell-1)}\right)\right\|_F^2$$

$$\le \gamma_\ell \|\boldsymbol{W}_\ell\|_F^2 \left\|\hat{\boldsymbol{Y}}^{(\ell-1)} - \boldsymbol{Y}^{(\ell-1)}\right\|_F^2$$

$$= \gamma_\ell \left\|\hat{\boldsymbol{Y}}^{(\ell-1)} - \boldsymbol{Y}^{(\ell-1)}\right\|_F^2. \tag{S.26}$$

Based on Proposition 1, the outcome of the first layer obeys $\|\hat{\boldsymbol{Y}}^{(1)} - \boldsymbol{Y}^{(1)}\|_F^2 \le \epsilon_1^2$, which together with (S.26) confirm the overall discrepancy.

## 4.4 Proof of Proposition 2

Since program (9) has only affine inequality constraints, the Karush–Kuhn–Tucker (KKT) optimality conditions give us necessary and sufficient conditions for an optimal solution. The pair $(\boldsymbol{w}^*, \boldsymbol{s}^*)$ is optimal if and only if there exists $\boldsymbol{\eta} \in \mathbb{R}^{n_2}$ and $\boldsymbol{\nu}$ such that

$$\eta_k s_k^* = 0, \quad k = 1, \ldots, n_2,$$
$$\eta_k \geq 0, \quad k = 1, \ldots, n_2,$$
$$\tilde{\boldsymbol{X}}^\top \boldsymbol{\nu} \in \begin{bmatrix} \partial \|\boldsymbol{w}^*\|_1 \\ \boldsymbol{\eta} \end{bmatrix}.$$

Above, $\partial \|\boldsymbol{w}^*\|_1$ denotes the subgradient of the $\ell_1$ norm evaluated at $\boldsymbol{w}^*$; the last expression above means that the first $n_1$ entries of $\tilde{\boldsymbol{X}}^\top \boldsymbol{\nu}$ must match the sign of $w_\ell^*$ for indexes $\ell$ with $w_\ell^* \neq 0$, and must have magnitude no greater than 1 for indexes $\ell$ with $w_\ell^* = 0$. The existence of such $(\boldsymbol{\eta}, \boldsymbol{\nu})$ is compatible with the existence of a $\boldsymbol{\Lambda}$ meeting the conditions in (10), by taking $\boldsymbol{\Lambda} = \tilde{\boldsymbol{X}}^\top \boldsymbol{\nu}$.

We now argue the conditions for uniqueness. Let $\boldsymbol{w}^*, \boldsymbol{s}^*, \boldsymbol{\Lambda}$ be as above. Suppose $(\boldsymbol{w}', \boldsymbol{s}')$ is a feasible point with $\|\boldsymbol{w}'\|_1 = \|\boldsymbol{w}^*\|_1$. Since $\boldsymbol{\Lambda}$ is in the row space of $\tilde{\boldsymbol{X}}$, we know that

$$\boldsymbol{\Lambda}^\top \begin{bmatrix} \boldsymbol{w}^* - \boldsymbol{w}' \\ \boldsymbol{s}^* - \boldsymbol{s}' \end{bmatrix} = 0,$$

and since $\boldsymbol{\Lambda}^\top [\boldsymbol{w}^*; \boldsymbol{s}^*] = \|\boldsymbol{w}^*\|_1$, we must also have $\boldsymbol{\Lambda}^\top [\boldsymbol{w}'; \boldsymbol{s}'] = \|\boldsymbol{w}^*\|_1$. Therefore by the properties stated in (10), the support (set of nonzero entries) $\tilde{\Gamma}$ of $[\boldsymbol{w}^*; \boldsymbol{s}^*]$ and $[\boldsymbol{w}'; \boldsymbol{s}']$ must be the same. Since these points obey the same equality constraints in the program (9), and $\tilde{\boldsymbol{X}}_{:,\tilde{\Gamma}}$ has full column rank, it must be true that $[\boldsymbol{w}'; \boldsymbol{s}'] = [\boldsymbol{w}^*; \boldsymbol{s}^*]$.

## 4.5 Proof of Theorem 3

For a more convenient notation we use $\Gamma = \operatorname{supp} \boldsymbol{w}^*$. Also, in all the formulations, sub-matrix selection precedes the transpose operation, e.g., $\boldsymbol{X}_{:,\Omega}^\top = (\boldsymbol{X}_{:,\Omega})^\top$.

Clearly since the samples are random Gaussians, with probability one the set $\{p : \boldsymbol{X}_{:,p}^\top \boldsymbol{w}^* = 0\}$ is an empty set, and following the notation in (9) and (10), $\operatorname{supp}^c \boldsymbol{s}^* = \varnothing$. Also, with reference to the setup in Proposition 2

$$\tilde{\boldsymbol{X}} = \begin{bmatrix} \boldsymbol{X}_{:,\Omega}^\top & \boldsymbol{0} \\ \boldsymbol{X}_{:,\Omega^c}^\top & -\boldsymbol{I} \end{bmatrix}.$$

To establish the full column rank property of $\tilde{\boldsymbol{X}}_{:,\tilde{\Gamma}}$ for $\tilde{\Gamma} = \operatorname{supp} \boldsymbol{w}^* \cup \{N + \operatorname{supp} \boldsymbol{s}^*\}$, we only

need to show that $\boldsymbol{X}_{\Gamma,\Omega}$ is of full row rank (thanks to the identity block in $\tilde{\boldsymbol{X}}$). Also, to satisfy the equality requirements in (10), we need to find a vector $\boldsymbol{\xi}$ such that

$$\begin{bmatrix} \boldsymbol{X}_{\Gamma,\Omega} & \boldsymbol{X}_{\Gamma,\Omega^c} \\ \boldsymbol{0} & -\boldsymbol{I} \end{bmatrix} \begin{bmatrix} \boldsymbol{\xi}_{\Omega} \\ \boldsymbol{\xi}_{\Omega^c} \end{bmatrix} = \begin{bmatrix} \mathrm{sign}(\boldsymbol{w}_{\Gamma}^*) \\ \boldsymbol{0} \end{bmatrix}. \tag{S.27}$$

This equation trivially yields $\boldsymbol{\xi}_{\Omega^c} = \boldsymbol{0}$. In the remainder of the proof we will show that when $P$ is sufficiently larger than $|\Gamma| = s$, the smallest eigenvalue of $\boldsymbol{X}_{\Gamma,\Omega}\boldsymbol{X}_{\Gamma,\Omega}^\top$ is bounded away from zero (which automatically establishes the full row rank property for $\boldsymbol{X}_{\Gamma,\Omega}$). Also, based on such property, we can select $\boldsymbol{\xi}_{\Omega}$ to be the least squares solution

$$\boldsymbol{\xi}_{\Omega} \triangleq \boldsymbol{X}_{\Gamma,\Omega}^\top \left( \boldsymbol{X}_{\Gamma,\Omega}\boldsymbol{X}_{\Gamma,\Omega}^\top \right)^{-1} \mathrm{sign}(\boldsymbol{w}_{\Gamma}^*), \tag{S.28}$$

which satisfies the equality condition in (S.27). To verify the conditions stated in (10) and complete the proof, we will show that when $P$ is sufficiently large, with high probability $\|\boldsymbol{X}_{\Gamma^c,\Omega}\boldsymbol{\xi}_{\Omega}\|_\infty < 1$. We do this by bounding the probability of failure via the inequality

$$\mathbb{P}\left\{\|\boldsymbol{X}_{\Gamma^c,\Omega}\boldsymbol{\xi}_{\Omega}\|_\infty \geq 1\right\} \leq \mathbb{P}\left\{\|\boldsymbol{X}_{\Gamma^c,\Omega}\boldsymbol{\xi}_{\Omega}\|_\infty \geq 1 \mid \|\boldsymbol{\xi}_{\Omega}\| \leq \tau\right\} + \mathbb{P}\left\{\|\boldsymbol{\xi}_{\Omega}\| > \tau)\right\}, \tag{S.29}$$

for some positive $\tau$, which is a simple consequence of

$$\mathbb{P}\left\{\mathcal{E}_1\right\} = \mathbb{P}\left\{\mathcal{E}_1|\mathcal{E}_2\right\}\mathbb{P}\left\{\mathcal{E}_2\right\} + \mathbb{P}\left\{\mathcal{E}_1|\mathcal{E}_2^c\right\}\mathbb{P}\left\{\mathcal{E}_2^c\right\} \leq \mathbb{P}\left\{\mathcal{E}_1|\mathcal{E}_2\right\} + \mathbb{P}\left\{\mathcal{E}_2^c\right\},$$

generally holding for two event $\mathcal{E}_1$ and $\mathcal{E}_2$. Without the filtering of the $\Omega$ set, standard concentration bounds on the least squares solution can help establishing the unique optimality conditions (e.g., see [3]). Here also, we proceed by bounding each term on the right hand side of (S.29) individually, while the bounding process requires taking a different path because of the dependence $\Omega$ brings to the resulting random matrices.

- **Step 1. Bounding $\mathbb{P}\{\|\boldsymbol{\xi}_{\Omega}\| > \tau\}$:**

By the construction of $\boldsymbol{\xi}_{\Omega}$ in (S.28), clearly

$$\|\boldsymbol{\xi}_{\Omega}\|^2 = \mathrm{sign}(\boldsymbol{w}_{\Gamma}^*)^\top \left( \boldsymbol{X}_{\Gamma,\Omega}\boldsymbol{X}_{\Gamma,\Omega}^\top \right)^{-1} \mathrm{sign}(\boldsymbol{w}_{\Gamma}^*). \tag{S.30}$$

Technically speaking, to bound the expression $\boldsymbol{x}^\top \boldsymbol{A}\boldsymbol{x}$, where $\boldsymbol{x}$ is a fixed vector and $\boldsymbol{A}$ is a self adjoint random matrix, we normally need the entries of $\boldsymbol{A}$ to be independent of the elements in $\boldsymbol{x}$. While such independence does not hold in (S.30) (because of the dependence of $\Omega$ to the entries of $\boldsymbol{w}_{\Gamma}^*$), we are still able to proceed with bounding by rewriting $\mathrm{sign}(\boldsymbol{w}_{\Gamma}^*) = \boldsymbol{\Lambda}_{\boldsymbol{w}^*}\boldsymbol{1}$, where

$$\boldsymbol{\Lambda}_{\boldsymbol{w}^*} = \mathrm{diag}\left(\mathrm{sign}(\boldsymbol{w}_{\Gamma}^*)\right).$$

Taking into account the facts that $\boldsymbol{\Lambda}_{\boldsymbol{w}^*} = \boldsymbol{\Lambda}_{\boldsymbol{w}^*}^{-1}$ and $w_n^* \neq 0$ for $n \in \Gamma$, we have

$$\|\boldsymbol{\xi}_\Omega\|^2 = \mathbf{1}^\top \left( \boldsymbol{\Lambda}_{\boldsymbol{w}^*} \boldsymbol{X}_{\Gamma,\Omega} \boldsymbol{X}_{\Gamma,\Omega}^\top \boldsymbol{\Lambda}_{\boldsymbol{w}^*} \right)^{-1} \mathbf{1}, \tag{S.31}$$

where now the matrix and vector independence is maintained. The special structure of $\boldsymbol{\Lambda}_{\boldsymbol{w}^*}$ does not cause a change in the eigenvalues and

$$\mathrm{eig} \left\{ \boldsymbol{\Lambda}_{\boldsymbol{w}^*} \boldsymbol{X}_{\Gamma,\Omega} \boldsymbol{X}_{\Gamma,\Omega}^\top \boldsymbol{\Lambda}_{\boldsymbol{w}^*} \right\} = \mathrm{eig} \left\{ \boldsymbol{X}_{\Gamma,\Omega} \boldsymbol{X}_{\Gamma,\Omega}^\top \right\},$$

where $\mathrm{eig}\{.\}$ denotes the set of eigenvalues. Now conditioned on $\boldsymbol{X}_{\Gamma,\Omega} \boldsymbol{X}_{\Gamma,\Omega}^\top \succ \mathbf{0}$, we can bound the magnitude of $\boldsymbol{\xi}_\Omega$ as

$$\begin{aligned}
\|\boldsymbol{\xi}_\Omega\|^2 &= \mathbf{1}^\top \left( \boldsymbol{\Lambda}_{\boldsymbol{w}^*} \boldsymbol{X}_{\Gamma,\Omega} \boldsymbol{X}_{\Gamma,\Omega}^\top \boldsymbol{\Lambda}_{\boldsymbol{w}^*} \right)^{-1} \mathbf{1} \\
&\leq \lambda_{\max} \left( \left( \boldsymbol{\Lambda}_{\boldsymbol{w}^*} \boldsymbol{X}_{\Gamma,\Omega} \boldsymbol{X}_{\Gamma,\Omega}^\top \boldsymbol{\Lambda}_{\boldsymbol{w}^*} \right)^{-1} \right) \mathbf{1}^\top \mathbf{1} \\
&= s \left( \lambda_{\min} \left( \boldsymbol{X}_{\Gamma,\Omega} \boldsymbol{X}_{\Gamma,\Omega}^\top \right) \right)^{-1}, \tag{S.32}
\end{aligned}$$

where $\lambda_{\max}$ and $\lambda_{\min}$ denote the maximum and minimum eigenvalues. To lower bound $\lambda_{\min} \left( \boldsymbol{X}_{\Gamma,\Omega} \boldsymbol{X}_{\Gamma,\Omega}^\top \right)$, we focus on the matrix eigenvalue results associated with the sum of random matrices. For this purpose, consider the independent sequence of random vectors $\{\boldsymbol{x}_p\}_{p=1}^P$, where each vector contains i.i.d standard normal entries. We are basically interested in concentration bounds for

$$\lambda_{\min} \left( \sum_{p \,:\, \boldsymbol{x}_p^\top \boldsymbol{w}^* > 0} \boldsymbol{x}_p \boldsymbol{x}_p^\top \right). \tag{S.33}$$

When the summands are independent self adjoint random matrices, we can use standard Bernstein type inequalities to bound the minimum or maximum eigenvalues [4]. However, as the summands in (S.33) are dependent in the sense that they all obey $\boldsymbol{x}_p^\top \boldsymbol{w}^* > 0$, such results are not directly applicable. To establish the independence, we can look into an equivalent formulation of (S.33) as

$$\lambda_{\min} \left( \sum_{p=1}^P \boldsymbol{x}_p \boldsymbol{x}_p^\top \right), \tag{S.34}$$

where $\boldsymbol{x}_p$ are independently drawn from the distribution

$$g_{\boldsymbol{X}}(\boldsymbol{x}) = \begin{cases} \frac{1}{\sqrt{(2\pi)^s}} \exp\left(-\frac{1}{2}\boldsymbol{x}^\top \boldsymbol{x}\right) & \boldsymbol{x}^\top \boldsymbol{w}^* > 0 \\ \frac{1}{2}\delta_D(\boldsymbol{x}) & \boldsymbol{x}^\top \boldsymbol{w}^* \leq 0 \end{cases}.$$

Here, $\delta_D(\boldsymbol{x}) = \prod_{i=1}^s \delta_D(x_i)$ denotes the $s$-dimensional Dirac delta function, and is probabilistically in charge of returning a zero vector in half of the draws. We are now theoretically able to apply the following result, brought from [4], to bound the smallest eigenvalue:

**Theorem S.1.** *(Matrix Bernstein[1]) Consider a finite sequence $\boldsymbol{Z}_p$ of independent, random, self-adjoint matrices with dimension $s$. Assume that each random matrix satisfies*

$$\mathbb{E}(\boldsymbol{Z}_p) = \boldsymbol{0}, \quad and \quad \lambda_{\min}(\boldsymbol{Z}_p) \geq R \quad almost\ surely.$$

*Then, for all $t \leq 0$,*

$$\mathbb{P}\left\{\lambda_{\min}\left(\sum_p \boldsymbol{Z}_p\right) \leq t\right\} \leq s \exp\left(\frac{-t^2}{2\sigma^2 + 2Rt/3}\right),$$

*where $\sigma^2 = \|\sum_p \mathbb{E}(\boldsymbol{Z}_p^2)\|$.*

To more conveniently apply Theorem S.1, we can use a change of variable which markedly simplifies the moment calculations required for the Bernstein inequality. For this purpose, consider $\boldsymbol{R}$ to be a rotation matrix which maps $\boldsymbol{w}^*$ to the first canonical basis $[1, 0, \cdots, 0]^\top \in \mathbb{R}^s$. Since

$$\mathrm{eig}\left\{\sum_{p=1}^{P} \boldsymbol{x}_p \boldsymbol{x}_p^\top\right\} = \mathrm{eig}\left\{\sum_{p=1}^{P} \boldsymbol{R}\boldsymbol{x}_p \boldsymbol{x}_p^\top \boldsymbol{R}^\top\right\}, \tag{S.35}$$

we can focus on random vectors $\boldsymbol{u}_p = \boldsymbol{R}\boldsymbol{x}_p$ which follow the simpler distribution

$$g_{\boldsymbol{U}}(\boldsymbol{u}) \triangleq \begin{cases} \frac{1}{\sqrt{(2\pi)^s}} \exp\left(-\frac{1}{2}\boldsymbol{u}^\top \boldsymbol{u}\right) & u_1 > 0 \\ \frac{1}{2}\delta_D(\boldsymbol{u}) & u_1 \leq 0 \end{cases}. \tag{S.36}$$

Here, $u_1$ denotes the first entry of $\boldsymbol{u}$, and we used the basic property $\boldsymbol{R}^{-1} = \boldsymbol{R}^\top$ along with the rotation invariance of the Dirac delta function to derive $g_{\boldsymbol{U}}(\boldsymbol{u})$ from $g_{\boldsymbol{X}}(\boldsymbol{x})$. Using the Bernstein inequality, we can now summarize everything as the following concentration result (proved later in the section):

**Proposition S.2.** *Consider a sequence of independent $\{\boldsymbol{u}_p\}_{p=1}^{P}$ vectors of length $s$, where each vector is drawn from the distribution $g_{\boldsymbol{U}}(\boldsymbol{u})$ in (S.36). For all $t \leq 0$,*

$$\mathbb{P}\left\{\lambda_{\min}\left(\sum_{p=1}^{P} \boldsymbol{u}_p \boldsymbol{u}_p^\top\right) \leq \frac{P}{2} + t\right\} \leq s \exp\left(\frac{-t^2}{P(s + 3/2) - t/3}\right). \tag{S.37}$$

Combining the lower bound in (S.32) with the concentration result (S.37) certify that when $P + 2t > 0$ and $t \leq 0$,

$$\mathbb{P}\left\{\|\boldsymbol{\xi}_\Omega\| \geq \sqrt{\frac{2s}{P + 2t}}\right\} \leq s \exp\left(\frac{-t^2}{P(s + 3/2) - t/3}\right). \tag{S.38}$$

- **Step 2. Bounding** $\mathbb{P}\{\|\boldsymbol{X}_{\Gamma^c,\Omega}\boldsymbol{\xi}_\Omega\|_\infty \geq 1 \mid \|\boldsymbol{\xi}_\Omega\| \leq \tau\}$:

Considering the conditioned event $\{\|\boldsymbol{X}_{\Gamma^c,\Omega}\boldsymbol{\xi}_\Omega\|_\infty \geq 1 \mid \|\boldsymbol{\xi}_\Omega\| \leq \tau\}$, we note that the set $\Omega$ is constructed by selecting columns of $\boldsymbol{X}$ that satisfy $\boldsymbol{X}_{:,p}^\top \boldsymbol{w}^* > 0$. However, since $\boldsymbol{w}_{\Gamma^c}^* = \boldsymbol{0}$, the index set $\Omega$, technically corresponds to the columns $p$ where $\boldsymbol{X}_{\Gamma,p}^\top \boldsymbol{w}_\Gamma^* > 0$. In other words, none of the entries of the sub-matrix $\boldsymbol{X}_{\Gamma^c,:}$ contribute to the selection of $\Omega$. Noting this, conditioned on given $\boldsymbol{\xi}_\Omega$, the entries of the vector $\boldsymbol{X}_{\Gamma^c,\Omega}\boldsymbol{\xi}_\Omega$ are i.i.d random variables distributed as $\mathcal{N}(0, \|\boldsymbol{\xi}_\Omega\|^2)$ and

$$\mathbb{P}\{\|\boldsymbol{X}_{\Gamma^c,\Omega}\boldsymbol{\xi}_\Omega\|_\infty \geq 1 \mid \|\boldsymbol{\xi}_\Omega\| \leq \tau\} = \mathbb{P}\left\{\bigcup_{n=1}^{|\Gamma^c|} |z_n| \geq \frac{1}{\|\boldsymbol{\xi}_\Omega\|} \;\middle|\; \|\boldsymbol{\xi}_\Omega\| \leq \tau\right\}, \tag{S.39}$$

where $\{z_n\}$ are i.i.d standard normals. Using the union bound and the basic inequality $\mathbb{P}\{|z_n| \geq a\} \leq \exp(-a^2/2)$ valid for $a \geq 0$, we get

$$\mathbb{P}\{\|\boldsymbol{X}_{\Gamma^c,\Omega}\boldsymbol{\xi}_\Omega\|_\infty \geq 1 \mid \|\boldsymbol{\xi}_\Omega\| \leq \tau\} \leq (N-s)\exp\left(-\frac{1}{2\tau^2}\right). \tag{S.40}$$

For $\tau = \sqrt{2s(P+2t)^{-1}}$ we can combine (S.40) and (S.38) with reference to (S.29) to get

$$\mathbb{P}\{\|\boldsymbol{X}_{\Gamma^c,\Omega}\boldsymbol{\xi}_\Omega\|_\infty \geq 1\} \leq s\exp\left(\frac{-t^2}{P(2s+1)-2t/3}\right) + (N-s)\exp\left(-\frac{P+2t}{4s}\right). \tag{S.41}$$

To select the free parameter $t$ we make the argument of the two exponentials equal to get

$$t^* = \frac{3s+4-\sqrt{45s^2+84s+25}}{12s+2}P,$$

for which the right hand side expression in (S.41) reduces to $N\exp(-(4s)^{-1}(P+2t^*))$. Based on the given value of $t^*$, it is easy to verify that for all $P \geq 0$ and $s \geq 1$, the conditions $t^* \leq 0$ and $P+2t^* > 0$ are satisfied. Moreover some basic algebra reveals that for all $P \geq 0$ and $s \geq 1$

$$-\frac{P+2t^*}{4s} \leq -\frac{P}{11s+7}.$$

Therefore, for $\mu > 1$, setting $P = (11s+7)\mu\log N$ guarantees that

$$\mathbb{P}\{\|\boldsymbol{X}_{\Gamma^c,\Omega}\boldsymbol{\xi}_\Omega\|_\infty \geq 1\} \leq N^{1-\mu}.$$

### 4.5.1 Proof of Proposition S.2

To use Theorem S.1, we focus on a sequence $\{\boldsymbol{Z}_p\}_{p=1}^P$ of the random matrices $\boldsymbol{Z} = \boldsymbol{u}\boldsymbol{u}^\top - \mathbb{E}(\boldsymbol{u}\boldsymbol{u}^\top)$. In all the steps discussed below, we need to calculate cross moments of the type $\mathbb{E}_g(u_1^{n_1}u_2^{n_2}\cdots u_s^{n_s})$ for $\boldsymbol{u} = [u_i] \in \mathbb{R}^s$ distributed as

$$g_U(\boldsymbol{u}) = \begin{cases} \frac{1}{\sqrt{(2\pi)^s}}\exp\left(-\frac{1}{2}\boldsymbol{u}^\top\boldsymbol{u}\right) & u_1 > 0 \\ \frac{1}{2}\delta_D(\boldsymbol{u}) & u_1 \leq 0 \end{cases}.$$

For the proposed distribution, the matrix of second order moments can be conveniently calculated as

$$\boldsymbol{D} = \mathbb{E}(\boldsymbol{u}\boldsymbol{u}^\top) = \frac{1}{2}\boldsymbol{I}.$$

The matrix $\boldsymbol{u}\boldsymbol{u}^\top$ is a rank one positive semidefinite matrix, which has only one nonzero eigenvalue. Using the Weyl's inequality we get

$$\lambda_{\min}(\boldsymbol{Z}) = \lambda_{\min}(\boldsymbol{u}\boldsymbol{u}^\top - \boldsymbol{D}) \geq \lambda_{\min}(\boldsymbol{u}\boldsymbol{u}^\top) + \lambda_{\min}(-\boldsymbol{D}) = -\frac{1}{2}. \tag{S.42}$$

Furthermore,

$$\mathbb{E}(\boldsymbol{Z}_p^2) = \mathbb{E}\left((\boldsymbol{u}\boldsymbol{u}^\top)^2\right) - \boldsymbol{D}^2,$$

for which we can calculate the expectation term as

$$\mathbb{E}\left((\boldsymbol{u}\boldsymbol{u}^\top)^2\right) = \frac{s+2}{2}\boldsymbol{I}.$$

Here, we used the following simple lemma:

**Lemma S.1.** *Given a random vector $\boldsymbol{u} = [u_i] \in \mathbb{R}^s$, with i.i.d entries $u_i \sim \mathcal{N}(0,1)$:*

$$\mathbb{E}\left((\boldsymbol{u}\boldsymbol{u}^\top)^2\right) = (s+2)\boldsymbol{I}.$$

It is now easy to observe that

$$\left\|\sum_{p=1}^{P}\mathbb{E}(\boldsymbol{Z}_p^2)\right\| = P\lambda_{\max}\left(\mathbb{E}\left((\boldsymbol{u}\boldsymbol{u}^\top)^2\right) - \boldsymbol{D}^2\right) = P\left(\frac{s+2}{2} - \frac{1}{4}\right) = \frac{P}{2}\left(s + \frac{3}{2}\right). \tag{S.43}$$

Now, using (S.42) and (S.43) we can apply Theorem S.1 to bound the smallest eigenvalue as

$$\forall t \leq 0: \quad \mathbb{P}\left\{\lambda_{\min}\left(\sum_{p=1}^{P}\boldsymbol{u}_p\boldsymbol{u}_p^\top - P\boldsymbol{D}\right) \leq t\right\} \leq s\exp\left(\frac{-t^2}{P(s+3/2) - t/3}\right). \tag{S.44}$$

Since $P\boldsymbol{D}$ is a multiple of the identity matrix, $\mathrm{eig}\{\sum_{p=1}^{P}\boldsymbol{u}_p\boldsymbol{u}_p^\top - P\boldsymbol{D}\} = \mathrm{eig}\{\sum_{p=1}^{P}\boldsymbol{u}_p\boldsymbol{u}_p^\top\} - P/2$ and therefore

$$\mathbb{P}\left\{\lambda_{\min}\left(\sum_{p=1}^{P}\boldsymbol{u}_p\boldsymbol{u}_p^\top\right) \leq \frac{P}{2} + t\right\} = \mathbb{P}\left\{\lambda_{\min}\left(\sum_{p=1}^{P}\boldsymbol{u}_p\boldsymbol{u}_p^\top - P\boldsymbol{D}\right) \leq t\right\} \tag{S.45}$$

which gives the probability mentioned in (S.37).

### 4.5.2 Proof of Lemma S.1

The $(i, j)$ element of the underlying matrix can be written as

$$\left((\boldsymbol{u}\boldsymbol{u}^\top)^2\right)_{i,j} = u_i u_j \sum_{k=1}^{s} u_k^2,$$

therefore,

$$\mathbb{E}\left(u_i u_j \sum_{k=1}^{s} u_k^2\right) = \left\{ \begin{array}{ll} 0 & i \neq j \\ \mathbb{E}\left(u_i^4 + u_i^2 \sum_{k \neq i} u_k^2\right) & i = j \end{array} \right. = \left\{ \begin{array}{ll} 0 & i \neq j \\ s + 2 & i = j \end{array} \right. . \tag{S.46}$$

Here, we used the facts that $\mathbb{E}(u_i^4) = 3$ and $\sum_{k \neq i} u_k^2 = s - 1$.

## Footnotes

[1]The original version of the theorem bounds the maximum eigenvalue. The present version can be easily derived using, $\lambda_{\min}(\boldsymbol{Z}) = -\lambda_{\max}(-\boldsymbol{Z})$ and $\mathbb{P}\{\lambda_{\min}(\sum_p \boldsymbol{Z}_p) \leq t\} = \mathbb{P}\{\lambda_{\max}(\sum_p -\boldsymbol{Z}_p) \geq -t\}$.