[Reviews · NeurIPS 2017]

Reviewer 1



The paper presents a technique to sparsify a deep ReLU neural network by solving a sequence of convex problems at each layer. The convex problem finds the sparsest set of weights that approximates the mapping from one layer to another. The ReLU nonlinearity is dealt with by treating the activated and deactivated cases as two separate sets of constraints in the optimization problem, thus, bringing convexity. Two variants are provided, one that considers each layer separately, and another that carries the approximation in the next layer optimization problem to give the chance to the next layer to counterbalance this error. In both cases, the authors provide bounds on the approximation error after sparsification. The paper is very clear and well-written. A number of theoretical results come in support to the proposed method. The experiments section shows results for various networks (fully connected, and convolutional) on the MNIST data. The results are compared with a baseline which consists of removing weights with smallest magnitude from the network. The authors observe that their method works robustly while the baseline methods lands in some plateau with very low accuracy. The baseline proposed here looks particularly weak. Setting a large number of weights to zero slashes the variance of the neuron preactivations, and the negative biases will tend to drive activations zero. Actually, in the HPTD paper, the authors obtain the best results by iteratively pruning the weights. A pruning ratio of 80% is quite low. Other approaches such as SqueezeNet have achieved 98% parameter reduction on image data without significant drop of performance.

Reviewer 2



In this paper the authors introduce an algorithm that sparsifies the weights of an already-trained neural network. The algorithm is a convex program based on l1-minimization and it applies layer-wise for layers with linear transform followed by ReLU. Theoretically the authors show that the outputs of the reduced network stay close to those of the original network; they also show that for Gaussian samples, with high probability the sparse minimizer is unique. They further apply their algorithm on MNIST dataset to show it is effective. The results in this paper are interesting and applicable to a wide range of trained networks. I checked the proofs and in my opinion they are correct and concise. The paper is well-written and the results of the experiments are illustrated clearly. On the negative side, the algorithm is only applicable for layers with ReLU nonlinearities (this is an important assumption in their proof), which rather limits the adaptability of the result. In the current version, reduction is only performed for fully-connected layers in CNNs. It will be very useful to have comparisons for reduced convolutional layers. The authors are urged to add results and comparisons for convolutional layers in the final version.

Reviewer 3



## Summary The authors address an important issue of compressing large neural networks and propose to solve a convex problem for each layer that sparsities the corresponding weight matrices. ## Major The paper is nicely written and the method seems sound–though I can’t comment on the correctness of the theorems, as I did not follow the proofs 100%. The biggest flaw is the restriction of the experiments to MNIST; we can’t say if the results scale to different data domains such as vision or typical regression data sets. ## Minor A reference to [1] is missing, in which an MDL inspired pruning method is proposed.